# A Lagrangian coherent eddy atlas for biogeochemical applications in the North Pacific Subtropical Gyre

Alexandra E. Jones-Kellett[1,2] and Michael J. Follows[1]

[1]Department of Earth, Atmospheric and Planetary Sciences, Massachusetts Institute of Technology, Cambridge, MA, USA
[2]Biology Department, Woods Hole Oceanographic Institution, Woods Hole, MA, USA

**Correspondence:** Alexandra E. Jones-Kellett (jonesae@mit.edu)

**Abstract.** Mesoscale eddies affect phytoplankton in several ways, including the horizontal dispersal and mixing of populations. Coherent eddies trap and contain fluid masses, while other eddies mix more freely with surrounding waters. To evaluate the role of lateral dispersal and trapping on the biogeochemical properties of eddies, we must accurately characterize their coherency. We employed a Lagrangian approach to identify materially coherent structures in remote sensing observations and developed a methodology to track them over their entire individual lifetimes. We provide an atlas of two decades of coherent eddies with an 8-day resolution in an intensely studied region of the North Pacific Subtropical Gyre (Jones-Kellett, 2023a, doi.org/10.5281/zenodo.8139149). The atlas was specifically designed to facilitate biogeochemical applications and was synchronized with available ocean color products. We identified coherent features using backward Lagrangian trajectories because the recent history of coherency (rather than the future coherency) will be most valuable for interpreting associated biogeochemical signatures. We compared the atlas of Lagrangian coherent eddies with an atlas of Eulerian eddies identified with the more conventionally used Sea Level Anomaly method. While 65% of tracked Sea Level Anomaly eddies are classified as coherent at some point in their lifetime, only 54% contain a Lagrangian coherent structure at any given time. We found similar variations in the temporal and spatial distributions of coherent and Sea Level Anomaly eddies. However, strictly coherent eddies have a clearer relationship between size and longevity and form distinct regional regimes based on polarity. We illustrate the potential of the coherent eddy atlas for biogeochemical applications by examining the relationship between bloom development and eddy evolution in a case study of a Hawaiian Lee cyclone.

## 1 Introduction

Mesoscale eddies are highly energetic features with horizontal scales of tens to hundreds of kilometers. They are typically generated by baroclinic instabilities, where the available potential energy of sloped density surfaces is converted into kinetic energy (reviewed by Ferrari and Wunsch, 2009). The mesoscale circulations are in approximate geostrophic balance in the horizontal and hydrostatic balance in the vertical. While small relative to major ocean gyres and current systems, eddies are extremely abundant and can propagate for weeks to months. Their associated vertical and submesoscale motions are known to influence local biogeochemistry, while their recirculating horizontal flows can be important for the retention and redistribution of marine tracers (reviewed by Mahadevan, 2016; McGillicuddy Jr, 2016; Lévy et al., 2018). These impacts on phytoplankton

populations, manifested as the pigment chlorophyll-*a* (chl-*a*), are observable from space. Disentangling the influence of biological processes and physical trapping is challenging because mesoscale eddies are not necessarily coherent from a Lagrangian perspective. Recently developed methods for identifying Lagrangian coherent eddy boundaries provide an opportunity to systematically evaluate eddy trapping and tease apart these influences.

In this study, we describe a methodology developed to track Lagrangian coherent structures en masse and present an atlas of two decades of mesoscale activity in the North Pacific Subtropical Gyre (NPSG). First, we review the use of remote sensing to identify mesoscale eddies and their biological signatures and describe previous work on Lagrangian eddy detection. In Section 2, we describe the algorithms employed to build the coherent eddy atlas. To this end, we synchronize the physical analysis to match the availability of remotely sensed ocean color products, use a backward-in-time Lagrangian trajectory analysis, and track features through time to examine eddy evolution. In Section 3, we show regional features of the eddy atlas, compare Eulerian and Lagrangian eddy characteristics, and present a case study following a localized chl-*a* bloom in a Hawaiian Lee cyclone. Finally, in Section 4, we end with a discussion of the results, analyses, and potential applications of the eddy atlas.

## 1.1 Mesoscale features in remotely sensed chlorophyll-*a*

The first ocean color observing satellite technology produced images from space that showed evidence of the control of mesoscale eddies on plankton patchiness (Gower et al., 1980). Since the 1990s, there has been continuous, global-scale remote sensing of surface chl-*a*. Simultaneous remote sensing of the physical environment provides an opportunity to identify patterns in the biogeochemical signatures of mesoscale circulations. There are several mechanisms by which mesoscale eddies can alter concentrations of chl-*a* including stirring, eddy-wind interaction, eddy pumping, and eddy trapping (reviewed by McGillicuddy Jr, 2016). The latter two are thought to enhance "monopole" chl-*a* anomalies, where elevated or depressed levels of chl-*a* are located in the core of an eddy. Satellite observations have revealed chl-*a* monopoles in eddies shed from the Gulf Stream and other western boundary currents (Waite et al., 2007; Gaube et al., 2013, 2014; He et al., 2022), the Agulhas rings (Lehahn et al., 2011; Villar et al., 2015), the North Pacific Ocean (Calil and Richards, 2010; Guo et al., 2017; He et al., 2022), the South Indian Ocean (Gaube et al., 2014), and the Southern Ocean (Dawson et al., 2018; Frenger et al., 2018). Disentangling the contributions of eddy trapping and pumping to chl-*a* monopoles remains a challenge, in part because dynamical trapping is difficult to quantify.

## 1.2 Eulerian eddy detection from satellite altimetry

Satellite altimeters measure the surface Sea Level Anomaly (SLA) from which the geostrophic flow is calculated by assuming a balance between the Coriolis force and the horizontal pressure gradient. Decades of these measurements from the Topex/Poseidon-ERS missions, Jason-Envisat, and more have enabled the identification and characterization of eddies globally. To determine dynamical eddy boundaries from satellite observations, early studies employed the Okubo-Weiss (OW) parameter (Okubo, 1970; Weiss, 1991) by demarking boundaries where the local vorticity dominates over the strain, calculated from the geostrophic velocity field (e.g., Isern-Fontanet et al., 2003; Chelton et al., 2007) (see Appendix A). However, this method

has high susceptibility to noise in the altimeter data, and thresholds to define boundaries are locally dependent, making it less ideal for global or large-scale eddy identification.

Chelton et al. (2011b) identified eddies globally by defining boundaries as closed contours surrounding local SLA extrema with better performance than the OW method. They found that most eddies identified directly from the SLA (here referred to as SLA eddies) are nonlinear, meaning the rotational fluid speed is greater than the translation speed. This characteristic distinguishes mesoscale eddies from Rossby waves, where a high ratio of the rotational to translation speeds enables eddies to potentially trap fluid (Flierl, 1981). Thus, Chelton et al. (2011b) concluded that most SLA eddies are "coherent". AVISO hosts and maintains a public global mesoscale eddy atlas (https://aviso.altimetry.fr/), increasing the accessibility and usage of the SLA method. The atlas provides trajectories that follow features over their lifetimes, which is particularly useful for studying the evolution of eddy influences on biogeochemical variables (Chelton et al., 2011a). Several derivative algorithms with various improvements have since been developed to generate public, global Eulerian eddy atlases from satellite altimetry (Mason et al., 2014; Faghmous et al., 2015; Tian et al., 2020; Pegliasco et al., 2022; Yang et al., 2023).

Regardless of algorithmic details, the OW and SLA methods are Eulerian. In other words, eddy boundaries are determined from satellite fields that are "frozen-in-time" reference frames. Although Eulerian methods are computationally inexpensive, they are ill-equipped to determine coherency because the boundaries of a materially invariant vortex must be frame-invariant (i.e., "objective"; reviewed by Haller, 2015). The nonlinearity parameter, previously used to suggest coherency, is a reference frame-dependent assessment when applied to Eulerian-detected eddies. Further, a study of satellite altimetry in the South China Sea found that the degree of leakage out of SLA eddies was independent of the nonlinearity parameter (Liu et al., 2022). An important distinction must be made that an SLA or OW eddy boundary may contain a trapping structure, but the Eulerian contour itself does not distinguish between fluid that is trapped and fluid that is actively mixing with surrounding waters.

### 1.3 Lagrangian coherency detection from satellite altimetry

Eddy boundaries derived from the SLA or OW parameter clearly fail tests of material invariance (e.g., Beron-Vera et al., 2013; Wang et al., 2015; Beron-Vera et al., 2019; Katsanoulis et al., 2020; Denes et al., 2022). And yet, many studies continue to assume that Eulerian eddy boundaries are coherent. To make such an assumption, Lagrangian coherence metrics can be used, but they are computationally expensive and rarely invoked for biophysical interpretations, with a few exceptions. He et al. (2022) found chl-*a* aggregations in coherent cyclonic eddies, and low chl-*a* in coherent anticyclones of the Northwest Pacific. Hernández-Carrasco et al. (2023) found that Lagrangian coherent eddies favor diatom growth in the oligotrophic Mediterranean Sea.

There is a history of investigation of the finite size and time Lyapunov exponents (FSLE/FTLE) and phytoplankton blooms. The FSLE and FTLE are Lagrangian coherent structure metrics that highlight stretching regions of the flow, including fronts and unstable manifolds (Pierrehumbert and Yang, 1992; Artale et al., 1997; Aurell et al., 1997). Case studies have identified FSLE (Lehahn et al., 2011; Calil and Richards, 2010; Guidi et al., 2012; Lehahn et al., 2014; Rousselet et al., 2018) or FTLE (Liu et al., 2018) streaks that surround chl-*a* anomalies at eddy cores. They show that Lagrangian coherent structures shape the spatiotemporal chl-*a* patterns at the ocean surface. However, the FSLE and FTLE metrics alone are not appropriate tools

to investigate eddy retention en masse because they simultaneously identify fronts and filaments that are not associated with eddies.

Several methods exist for Lagrangian coherent eddy detection (reviewed by Hadjighasem et al., 2017). One of which is the Lagrangian Averaged Vorticity Deviation (LAVD), an observer-independent method that monitors the vorticity of fluid parcels to reveal Rotationally Coherent Lagrangian Vortices (RCLVs; Haller et al., 2016). Here we define Lagrangian eddies (or RCLVs) using the LAVD method due to its objectivity, computational efficiency, and mathematical foundation describing material rotation. RCLVs are ubiquitous in satellite altimetry observations (Abernathey and Haller, 2018; Tian et al., 2022; Xia et al., 2022b; Liu and Abernathey, 2023), and their trapping properties have been validated with in situ trajectories of Argo floats (Liu and Abernathey, 2023) and drifters (Encinas-Bartos et al., 2022).

Comparisons of RCLVs and SLA eddies provide insight into the trapping properties of mesoscale features. In an idealized model of the North Pacific, Liu et al. (2019) distinguished three types of eddy features: non-overlapping SLA eddies, non-overlapping RCLVs, and overlapping eddies. Overlapping features are detected with both methods and non-overlapping are detected by one. RCLVs tend to be smaller and nested within the bounds of SLA eddies for overlapping features. Liu et al. (2019) found that the leakage rate of non-overlapping SLA eddies was higher than for overlapping eddies and that their dispersal properties were not significantly different from random ocean pieces of the same size. Satellite observations of the Northwest Pacific (Yuan and Hu, 2023) and the South China Sea (Liu et al., 2022) indicate that non-overlapping SLA eddies lose roughly 80% of their initial water over their lifetimes.

To interpret biogeochemical developments in eddies, it is useful to study eddy evolution by tracking features over the entirety of their lifespans as SLA eddy atlases do. Because the LAVD method calculates coherency over fixed time intervals, an equivalent RCLV atlas does not exist, which is currently a major shortcoming of the method (Liu and Abernathey, 2023). Evaluations of RCLV evolution have only been performed for case studies and over limited domains. For example, Vortmeyer-Kley et al. (2019) tracked short-lived RCLVs in a simulation of the Baltic Sea, Xia et al. (2022a) examined the details of Lagrangian coherency in Agulhas ring evolution, and others followed RCLVs to understand transport in the Caribbean Sea and the Gulf of Mexico (Andrade-Canto et al., 2020, 2022; Ntaganou et al., 2023). Nevertheless, the literature mentioned above does not provide methodologies that are well-suited for systematically tracking the evolution of a large number of RCLVs, as is characteristic of an eddy atlas.

Besides the direct applications for interpreting biogeochemical responses, there are several open questions that a Lagrangian coherent eddy atlas would address. Previous studies found that fewer and smaller structures maintain coherency for longer timescales (Abernathey and Haller, 2018; Xia et al., 2022b; Liu and Abernathey, 2023). However, the average lifespan of a coherent structure is unknown. How coherent structures change in size throughout their life is also unexplored. Furthermore, it is unclear whether eddies maintain the same coherent properties over their lifetimes or if they vary. For example, are non-overlapping RCLVs always non-overlapping, or do they overlap with SLA eddies at some point? Is there a life stage where eddies are most likely to be coherent? Comparing RCLV and SLA eddy atlases can answer these questions, revealing how the coherent properties of mesoscale features manifest in space and time.

This study aims to present and evaluate a publicly available RCLV atlas product that is useful for applications to characterize the biogeochemical response to eddy trapping. Phytoplankton populations grow on timescales of the order of days, and acclimation of pigments can be even faster. Thus a high temporal resolution mapping of eddy coherency is valuable to interpret biophysical interactions in eddies. Furthermore, biogeochemical states are a function of the physical environment in the recent past, rather than the future. Existing RCLV datasets detect features from satellite observations every 30 days or longer

using forward-in-time Lagrangian simulations (Abernathey and Haller, 2018; Tian et al., 2022; Xia et al., 2022b; Liu and Abernathey, 2023). Here we used backward-in-time Lagrangian trajectories to identify RCLVs every 8 days from 2000-2019 and developed an algorithm to track features over the entirety of their lifespans. This is the highest resolution of backward-in-time RCLVs measured en masse from satellite altimetry and the first RCLV atlas. We focused on the NPSG region that encompasses the Hawai'i Ocean Time-Series site at Station ALOHA (Karl and Church, 2019) and the locations of other major

biogeochemical sampling campaigns that targeted mesoscale features (Falkowski et al., 1991; Seki et al., 2001; Vaillancourt et al., 2003; Benitez-Nelson et al., 2007; Benitez-Nelson and McGillicuddy Jr, 2008; Zhang et al., 2021; Barone et al., 2022). The atlas provides a resource for retrospective analysis of in situ and remote observations and for contextual planning of future expeditions.

## 2    Materials and methods

### 2.1    Satellite data


Daily Level 4 satellite geostrophic current data from 2000-2019 were obtained from the $1/4°$ Version $008\_047$ gridded global ocean dataset distributed by CMEMS. This product uses the DUACS multi-mission altimeter data processing system, which combines data from all available altimetry satellites. The SLA is computed with respect to a 20-year mean, and geostrophic currents are derived from the SLA. We restricted the spatial domain of our analysis to 15-30°N and 180-230°W. This encom-

passes the eastern NPSG, excluding the ultra-oligotrophic waters to the west, the transition zone to the north (Glover et al., 1994), and the equatorial currents to the south.

For the case study in Section 3.4, 8-day average satellite chl-*a* data were obtained for 2010 from Version 6.0 of the Ocean Color Climate Change Initiative (OC-CCI) project. The product merges multiple ocean color sensor products (MERIS, SeaWiFS, MODIS-Aqua, and VIIRS) for improved coverage, with a nominal spatial resolution of $4\ km$ at the equator (Sathyen-

dranath et al., 2019).

### 2.2    Eulerian Sea Level Anomaly eddies

We used the OceanEddies MATLAB software (Faghmous et al., 2015) to detect eddies from satellite SLA and compare them with the RCLV atlas described in the following sections. The OceanEddies algorithm provides flexibility in parameter choices, allowing us to align the detection criteria in the Eulerian and Lagrangian eddy atlases as closely as possible. The algorithm

identifies an eddy boundary as the outermost closed contour containing a single SLA extremum. We required SLA contours

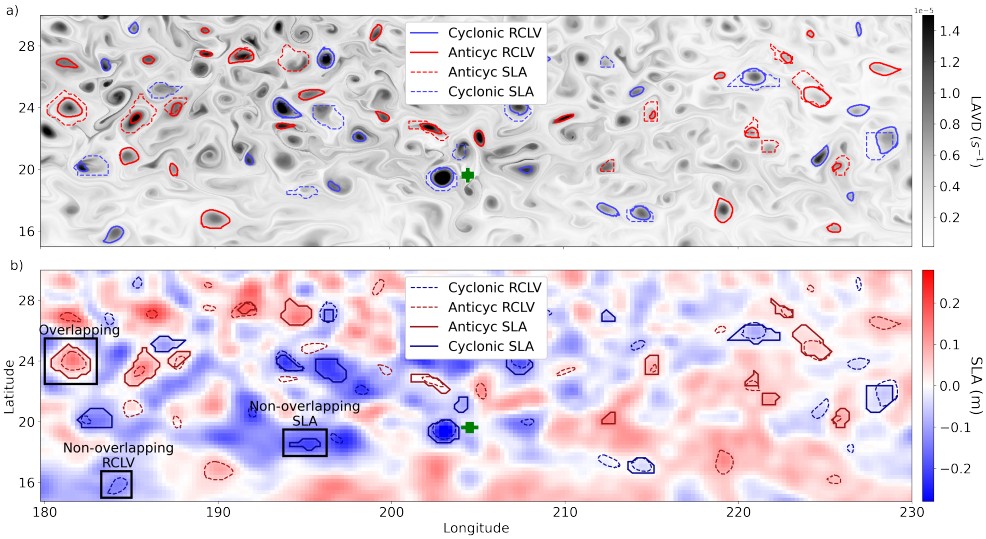

**Figure 1.** (a) 32-day backward-in-time LAVD field calculated from Lagrangian particles initialized on 23 April 2010. RCLV boundaries are plotted with solid lines. For comparison, the boundaries of SLA eddies that are at least 32 days old are plotted with dotted lines. (b) The SLA field on 23 April 2010. The black rectangles highlight cases of an overlapping eddy, a non-overlapping SLA eddy, and a non-overlapping RCLV.

to contain at least 12 grid cells to account for noise in the data and prevent spurious detection of small features. As a result, the smallest SLA eddy area in the dataset is $8,048\ km^2$, which has a radius of approximately $50\ km$. This is consistent with the Rossby radius of deformation in the region of interest (Chelton et al., 1998). Eddies were detected at a daily frequency concurrent with the temporal resolution of the satellite SLA product. To account for noise in the SLA and to prevent prematurely

"killing" an eddy, we set the eddy disappearance parameter to three days. For the subsequent analysis, we reduced the temporal resolution of the SLA product to an 8-day frequency to align with the RCLV dataset (described in Section 2.4). We also required eddies to have a minimum lifetime of 32 days and removed eddy contours younger than 32 days from the analysis, again to align with the RCLV dataset. Figure 1(b) illustrates an instantaneous satellite SLA field and the corresponding eddy features.

## 2.3   Lagrangian particle simulations

We conducted Lagrangian particle simulations by implementing a 4th-order Runge-Kutta advection scheme using Ocean-Parcels version 2.2.2, an open-source Python software (Delandmeter and van Sebille, 2019). Particle position $(x,y)$ was calculated from the geostrophic velocity $(u,v)$, such that

$$\frac{\delta x}{\delta t} = u, \frac{\delta y}{\delta t} = v \tag{1}$$

When seeding the spatial domain with a field of Lagrangian particles, the initial particle spacing must be smaller than the velocity grid size to accurately identify RCLVs (Abernathey and Haller, 2018). Accordingly, particles were initialized in an $8 \times 8$ grid within each $1/4°$ cell of the velocity grid, resulting in a particle initialization resolution of $1/32°$. This amounted to 768,000 Lagrangian particles in each simulation. We created a custom Kernel within OceanParcels to calculate the relative vorticity ($\zeta$) along the trajectories, where

$$\zeta = \frac{\partial v}{\partial x} - \frac{\partial u}{\partial y} \tag{2}$$

Particle location and relative vorticity were calculated every 20 minutes. This timestep was selected based on sensitivity tests of particle trajectory convergence. To reduce storage size, the data were output every six hours. We performed sensitivity tests to ensure that this did not reduce the quality of the LAVD field and the RCLV boundary detection. Lagrangian simulations were run backward in time for 32 days to describe the history of fluid parcels. RCLVs derived from backward simulations are coherent over the simulation time interval but need not maintain coherency into the future. The 32-day timescale was chosen to measure coherent features analogous to those in preceding literature (e.g., 30-day RCLVs detected by Abernathey and Haller (2018) and Liu and Abernathey (2023)) and to conveniently align with 8-day ocean color products.

Most previous RCLV studies ran Lagrangian simulations on temporally non-overlapping data (Abernathey and Haller, 2018; Liu et al., 2019; Sinha et al., 2019; Zhang et al., 2020; Tian et al., 2022; Xia et al., 2022b), meaning that subsequent simulations were initialized when the previous ended. This is less computationally expensive than running overlapping simulations, but they do not capture the evolution of the boundaries of a coherent structure at a high resolution. One could track particles between two non-overlapping simulations to interpolate RCLV boundaries, but this would prevent the structure from growing or shrinking for the timescale of the simulations (i.e., 32 days, in our case). Instead, we re-initialized the Lagrangian particles every 8 days to capture boundary evolution at a high resolution. We intentionally aligned the dates of initialization with the 8-day average OC-CCI ocean color products. This amounted to 920, 32-day simulations (3.7 Gb each, 3.4 Tb total) for the 20-year domain and 706,560,000 particle trajectories.

## 2.4 RCLV atlas

The code used to perform the Lagrangian simulations on CMEMS geostrophic velocity data with OceanParcels and to create the RCLV atlas presented here is available at github.com/lexi-jones/RCLVatlas. Additionally, the 20-year atlas of the NPSG is distributed by Simons CMAP at simonscmap.com/catalog/datasets/RCLV_atlas.

### 2.4.1 Boundary detection

The LAVD is a measure of local rotation experienced by a Lagrangian particle integrated over a time interval of interest. For a given fluid element $\mathbf{p}$,

$$LAVD_{t_0}^{t_1}(\mathbf{p}) = \frac{1}{t_1 - t_0} \int_{t_0}^{t_1} |\zeta(\boldsymbol{X}(\mathbf{p}, \tau); \tau) - \overline{\zeta(\tau)}| d\tau \tag{3}$$

where $X$ is a Lagrangian trajectory such that $X(\mathbf{p}, t)$ is the fluid element's position in space at time $t$, $\zeta(X(\mathbf{p}, t); t)$ is the relative vorticity of the fluid element at time $t$, $\overline{\zeta(t)}$ is the average relative vorticity across the spatial domain at time $t$, and $[t_0, t_1]$ is the temporal domain of the Lagrangian simulation. A Lagrangian particle with high LAVD experiences high magnitude relative vorticity along its trajectory over the period of interest. An RCLV is defined as a set of fluid elements that rotate around an eddy core in a manner analogous to rigid-body rotation (Haller et al., 2016). Fluid elements along a materially invariant boundary have the same average angular speed and vorticity, and thus equally large LAVD values. Accordingly, an RCLV is identified in the LAVD field as a closed contour surrounding a local maximum (Haller et al., 2016; Tarshish et al., 2018). The RCLV represents a feature that maintains coherency from $\tau = t_0$ to $\tau = t_1$.

We calculated the 32-day, backward-in-time LAVD for every Lagrangian particle in 920 simulations. The LAVD field is visualized by plotting the value for each particle at the initialization time and location (see Fig. 1(a)). Moving forward, we refer to the initialization dates of each of the LAVD fields as the "timesteps" of our dataset (each separated by eight days), since this is the frequency at which RCLVs boundaries were identified. The Python package *floater* was used to identify RCLVs from local LAVD maxima encompassed by closed contours (Tarshish et al., 2018). The algorithm has several free parameters that must be carefully chosen, as they affect what qualifies as an RCLV boundary. This includes minimum area, minimum distance between features, and the contour Convex Deficiency (CD).

Since the LAVD field has a finer spatial resolution than the SLA field, we can detect smaller RCLVs than SLA eddies. Moreover, RCLV radii tend to be half the length of SLA eddy radii (Abernathey and Haller, 2018; Liu and Abernathey, 2023). Assuming a $20\ km$ minimum radius, a circular feature of this size would have an area of $1257\ km^2$ or contain roughly 104 grid cells ($1/32°$). Therefore, we set the minimum RCLV area to 104 grid cells. The minimum RCLV radius of $20\ km$ is roughly equivalent to the length scale of six or more grid cells, so for the edge of two minimally sized RCLVs to be touching the distance between their centers will be 12 grid cells. Assuming that RCLV eddy centers are at least an "eddy distance" away from each other, we set the minimum distance between the local maxima of the LAVD field to 24 grid cells.

The CD is the ratio of the area between the contour and its convex hull to the area enclosed by the contour (Haller et al., 2016). The convex hull is the closed curve with the shortest perimeter containing the polygon. For example, a perfectly circular contour has $CD = 0$ because the convex hull is identical to the contour itself. Conversely, the area between a "wiggly" contour and its convex hull is higher and yields a high CD (Fig. 2(e)). In the context of the LAVD field, an RCLV contour with a low CD presumably contains a fluid structure with minimal filamentation. Choosing a threshold for the CD is somewhat arbitrary since the boundaries of two different vortices with the same CD can have differing degrees of dispersal. Tarshish et al. (2018) introduced a complementary metric to the CD called the Coherency Index (CI). The CI measures the spatial compactness of a set of Lagrangian particles over time, defined

$$CI = \frac{\sigma^2(0) - \sigma^2(t)}{\sigma^2(0)} \tag{4}$$

where $\sigma^2(t) = <|X(t) - <X(t)>|^2>$ is the variance of the particle positions at time $t$, and the $<>$ symbols indicate the mean. $<X(t)>$ is the mean position of all particles or the feature center. A negative CI indicates that the particle set spread apart over the time frame, whereas a positive CI indicates that it contracted.

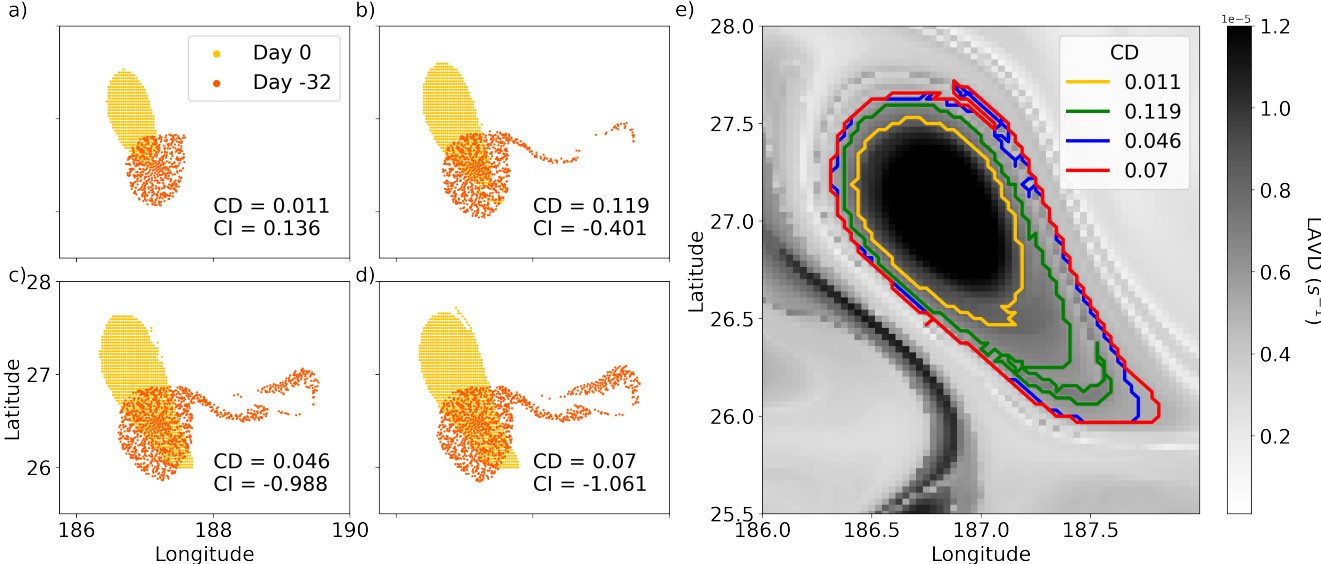

**Figure 2.** An RCLV vortex and its potential boundaries derived by varying the target CD in the floater algorithm. (a) Particles were initialized inside the contour with $CD = 0.011$ and advected backward in time for 32 days. The particles are yellow at the initialization, and orange at their locations after advection. (b) Particles initialized in the contour with $CD = 0.119$. (c) Particles initialized in the contour with $CD = 0.049$. (d) Particles initialized in the contour with $CD = 0.07$. (e) Contour output from different target CDs.

Tarshish et al. (2018) correlated CI with various choices of CD to select a threshold CD to define a coherent eddy. Informed by our sensitivity tests, we refined their method by searching for the largest contour surrounding a local maximum that satisfies the conditions $CD <= 0.03$ and $CI >= -0.5$. Because CD and CI are not linearly correlated (Fig. B1), we found that this is an appropriate, conservative measure to identify robust structures with limited filamentation. Figure 2 shows particle trajectories initialized inside four different potential contour boundaries of an RCLV. Only the smallest contour with target $CD = 0.01$ passed the criteria of $CD <= 0.03$ and $CI >= -0.5$.

In some rare cases, RCLV contours passed the CD and CI thresholds but did not encompass a single materially invariant vortex. The LAVD metric is derived from the absolute value of the relative vorticity, thus filamentations of opposite polarity could occur amongst particles in a supposed RCLV contour so long as they maintain integrated vorticity of a high magnitude. To exclude such behavior from our dataset, we ran sensitivity tests that determined the requirement for 85% of particles within a contour to have the same sign of vorticity at the beginning and end of the simulations. Thus to qualify as an RCLV, if at least 85% of particles in a feature had negative (positive) vorticity at the initialization of the simulation, 85% must also have had negative (positive) vorticity 32 days previously. This requirement is a more conservative identification of RCLVs than in previous literature, but one that we found necessary to identify robustly coherent features. We assigned RCLV polarity by the dominant sign of the vorticity of the particles inside of the boundaries, where cyclones (anticyclones) had a positive (negative) sign of relative vorticity.

**Table 1.** Summary of free parameters

| | |
|---|---|
| **SLA eddies** | |
| Minimum age | 32 days |
| Minimum area | 12 grid cells ($1/4°$) |
| Disappearance tolerance | 3 days |
| | |
| **Lagrangian simulations** | |
| Initial particle resolution | $1/32°$ |
| Particle timestep | 20 minutes |
| Output frequency | 6 hours |
| Run time | 32 days |
| Re-initialization frequency | 8 days |
| | |
| **RCLVs** | |
| Minimum area | 104 grid cells ($1/32°$) |
| Minimum distance between centers | 24 grid cells |
| Maximum CD | 0.03 |
| Minimum CI | -0.5 |
| Minimum vorticity consistency | 85% |

The RCLV parameters were selected to delineate boundaries that contain minimally filamentous vortices, as exhibited in Fig. 2. Depending on the scientific application, one might choose alternative criteria for eddy coherency or choose different parameter values when analyzing a different dataset. The parameter choices made for this study are summarized in Table 1.

### 2.4.2 RCLV tracking

On account of our strict thresholds for RCLV boundaries, following the advection pathway of a single particle was sufficient to track coherent water masses between the 8-day timesteps. To do so, we used the same Lagrangian trajectories that were simulated to compute the LAVD. For each unique coherent water mass, the algorithm proceeds as follows:

1. Starting at the last chronological timestep (on July 4th in Fig. 3), the RCLV is assigned a unique ID.

2. The Lagrangian particle corresponding with the LAVD local maximum is tracked backward in time (from July 4th to June 26th in Fig. 3). This represents the center of the rotating rigid body and is the particle most likely to remain inside of the vortex for the extent of its lifetime.

3. If the Lagrangian particle is advected into a contour in the previous chronological timestep (June 26th in Fig. 3), then that RCLV contour is also assigned the unifying ID.

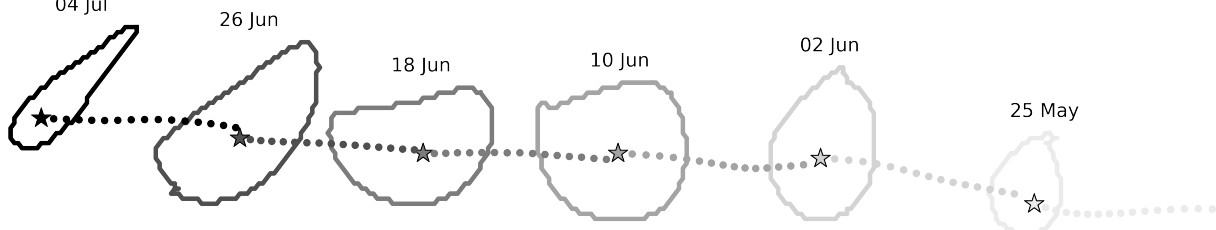

**Figure 3.** Illustration of the vortex tracking method for an RCLV transiting westward. The solid lines represent the RCLV contours every eight days, where the color corresponds with the eddy age (darkest = oldest). The stars indicate the location of the local maxima of the LAVD at the corresponding timestep, and the dots represent the backward trajectory of the Lagrangian particle to the previous timestep. The water mass developed into a coherent structure 32 days prior to May 25th and remained coherent until July 4th. The contours and particle trajectories were artificially spread apart here for visual purposes.

4. Repeat Steps 2. and 3. until the particle is not advected inside a contour in the previous chronological timestep (on May 25th in Fig. 3). This represents the RCLV birth, i.e., the first timestep that the eddy was coherent for at least 32 days.

5. The RCLV instances are assigned an age, with the first chronological contour called 32 days old (on May 25th in Fig. 3).

265     Although the feature in Fig. 3 contains the same core water mass from May 25th to July 4th, the vortex changes in size with time. Our method inherently allows the RCLV to have a fluctuating material belt on the eddy periphery Xia et al. (2022a) because the contours are detected independently at each timestep. In other words, each contour contains fluid that was part of the RCLV for at least 32 days, but all of the fluid need not be part of the structure for the full lifespan of the tracked eddy. Each instantaneous contour represents the coherent boundary at that time, but in most cases, only an inner core is part of
270     the coherent structure for the entirety of the RCLV lifetime. This method captures the RCLV evolution that a phytoplankton population would realistically be subject to.

### 2.4.3   Disappearing RCLVs

We found some cases where two RCLV contour sets encompass the same fluid mass, but the eddy "disappears" for one or more timesteps. This could occur if a coherent structure reduced in size below the minimum threshold, or the RCLV contour
275     at the skipped timestep had a CD or CI just outside of the parameter limits. Other possibilities include an eddy splitting or merging event. To avoid prematurely terminating an eddy, our algorithm allows RCLVs to disappear for up to two timesteps to accurately attribute RCLV age and examine feature evolution. To interpolate through this disappearance, the Lagrangian trajectories of particles inside of the first chronological contour following the skipped timestep(s) are tracked backward in time. A polygon is drawn around the mass of particles at each missing timestep, and all contours are linked with a unifying
280     RCLV ID number. 9.4% of contours in our published dataset were interpolated in this manner and are flagged.

**Table 2.** Number of eddies from 2000-2019

|  | All | Anticyclonic | Cyclonic |
|---|---|---|---|
| RCLV instances | 40644 | 18200 | 22444 |
| Tracked RCLV IDs | 11900 | 5615 | 6285 |
| Daily average number of RCLVs | 44.2 | 19.8 | 24.4 |
| SLA eddy instances | 28482 | 13395 | 15087 |
| Tracked SLA eddy IDs | 5990 | 2858 | 3132 |
| Daily average number of SLA eddies | 31.0 | 14.6 | 16.4 |

Only includes eddies that are 32 days old or more with an 8-day detection frequency.

## 3 Results

In this section, we present a summary and key features of the 20-year RCLV atlas of the NPSG and compare with an SLA eddy atlas. Figure 1 portrays a comparable density of RCLVs and SLA eddies in the region and shows that numerous features were identified with both methods, though many were only identified in one. Video S1 displays the dynamic evolution of the LAVD, SLA, and eddy boundaries over one year. The Video illustrates that most eddies transit westward in the domain such that their displacement is influenced by the large-scale flow.

The number of eddies in the RCLV and SLA atlases are summarized in Table 2. The RCLV atlas contains 40,644 contours, with 11,900 unique IDs associated with features tracked within the bounds of the domain. Daily-resolved atlases would have 8 times as many contours as presented here, but the same number of IDs/tracked features. On any given day, there are on average 44 RCLVs. SLA eddies are less frequent, with 5,990 unique IDs and a mean of 31 per day. The lower frequency of SLA eddies compared to RCLVs is in part due to the stricter size thresholds. However, we cannot accurately detect smaller Eulerian eddies with the current spatial resolution of remotely sensed SLA ($1/4°$). Cyclonic eddies are more frequent than anticyclonic eddies in both datasets, although this bias varies regionally (see Section 3.3).

The median lifespan for tracked RCLVs is 40 days, compared to 55 days for SLA eddies. SLA eddies tend to have a longer lifespan than RCLVs except for in the Lee of Hawai'i. Even so, the Lee Eddies are the longest-lived features in both datasets. Eddies in the southeast quadrant can also have long lifespans (upwards of 100 days) as they transit westward toward the south of the islands. In the remainder of the domain, RCLVs do not typically survive beyond 64 days. We did not find significant differences in eddy age between the polarities in either dataset.

We compared the co-occurrence of eddy contours in the SLA and RCLV atlases at each 8-day timestep, presented in Fig. 4. SLA eddy contours that do not co-occur with an RCLV are referred to as non-overlapping SLA eddies as opposed to co-occurring overlapping eddies (Liu et al., 2019). We found that 57% of cyclonic and 51% of anticyclonic SLA eddy contours overlap with an RCLV. Over the lifetimes of tracked SLA eddies, 67% of cyclones and 62% of anticyclones overlap with RCLVs at some point. This suggests that while roughly two-thirds of SLA eddies trap material during their lifetime, only half

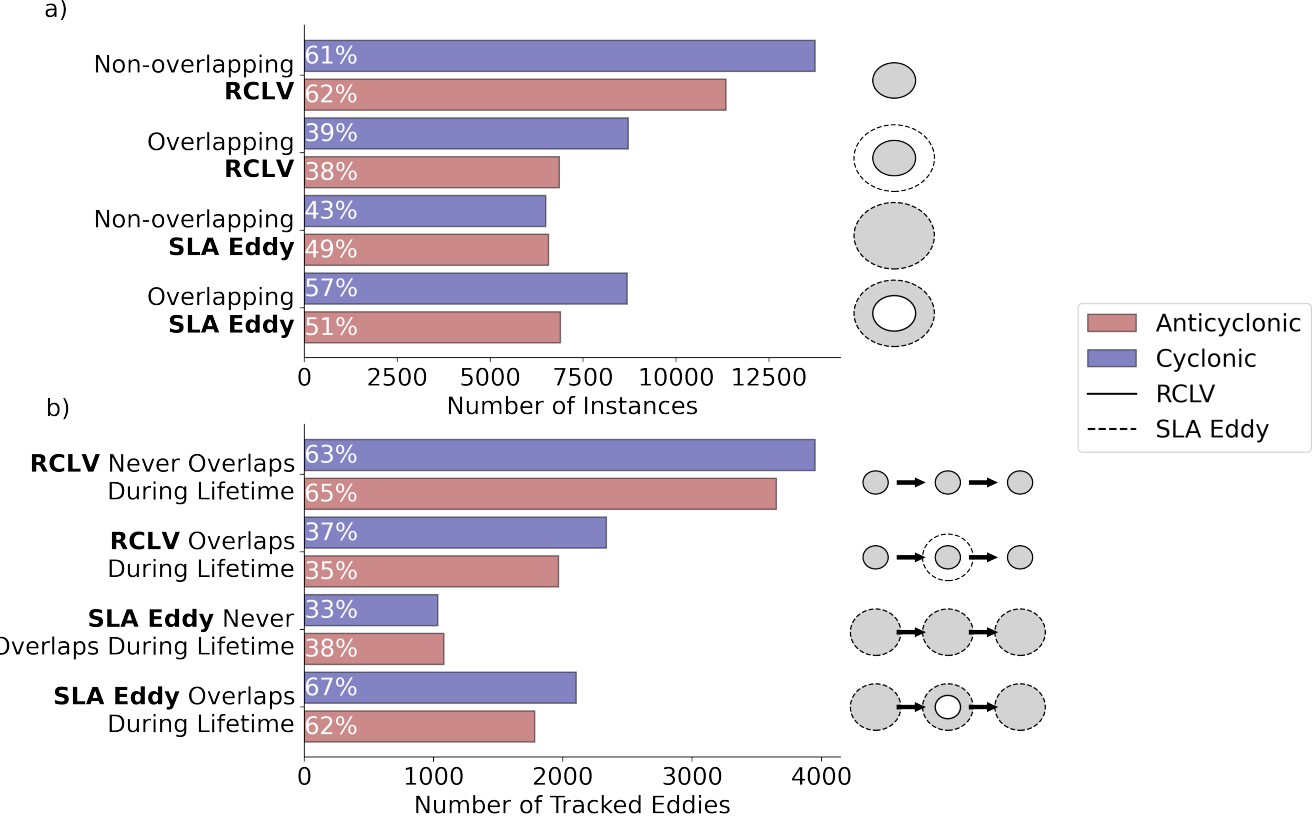

**Figure 4.** (a) The number of eddy contours that are concurrent (overlapping) and not concurrent (non-overlapping) with contours in the other atlas. The percentages of eddies exhibiting the given characteristic compared to the total number of eddies of each polarity are overlaid in white. For overlapping eddies, the numbers are the same since this is a mutual metric, but the percentages of the total eddy counts differ. The schematics on the right-hand side show the referenced feature in gray. (b) The number of eddies that overlap and never overlap throughout the lifetimes of the tracked features.

have significant retentive properties at any given time. More RCLVs were identified than SLA eddies, thus 63% of cyclonic and 65% of anticyclonic RCLVs never overlap with SLA eddies.

Figure 5 shows an example of the difference in the retentiveness of the boundaries of an overlapping eddy. We initialized Lagrangian particles in an eddy on May 17th and advected them backward through time. At the initialization (Fig. 5(a)), the orange particles are inside the SLA eddy boundary but outside the RCLV, whereas the green particles are inside the RCLV. While the green particles were trapped in the eddy eight days prior (May 9th), 62% of the orange particles were outside of the eddy (Fig. 5(b)). We found that fewer of the orange particles remained in the eddy the longer we advected the particles backward in time, and the farther away from the eddy they originated. This demonstrates the significant lateral exchange that occurs with surrounding waters in the zone of overlapping eddies between the RCLV and SLA eddy boundaries.

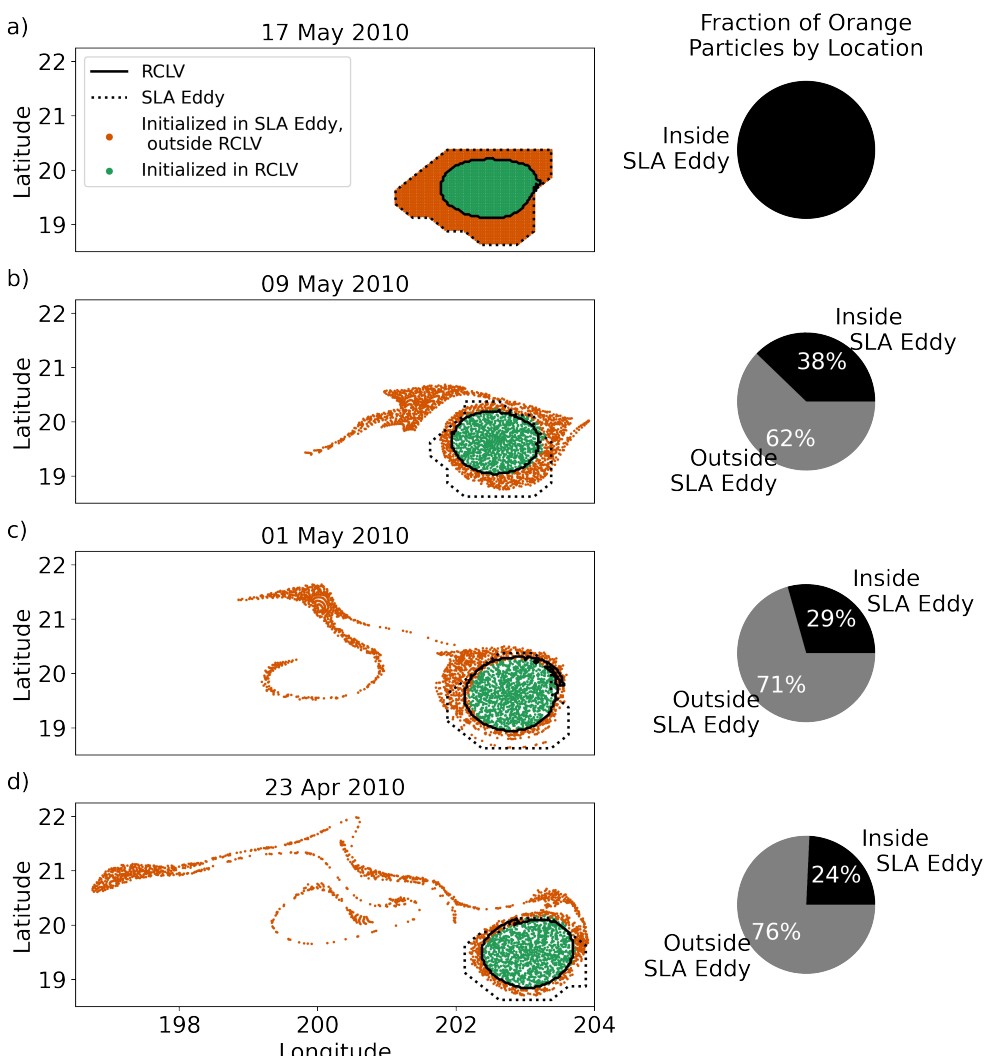

**Figure 5.** (a) Lagrangian particles initialized in an overlapping eddy (RCLV ID 6724) on 17 May 2010. (b) Particle locations backward-in-time on 9 May 2010, (c) 1 May 2010, and (d) 23 April 2010. The pie charts show the fraction of orange particles inside or outside the SLA eddy boundaries.

## 3.1 Temporal variability

RCLVs and SLA eddies have similar annual and interannual variability, as shown in Fig. 6, suggesting common underlying mechanisms. The monthly average eddy frequency data that underlies Fig. 6(b) is provided in Appendix Table C1. Daily eddy frequency peaks in June and July and is lowest in December and January, consistent with previously noted summer peaks of eddy kinetic energy (EKE) in subtropical gyres (Zhai et al., 2008). Nevertheless, by only including eddy occurrences that are

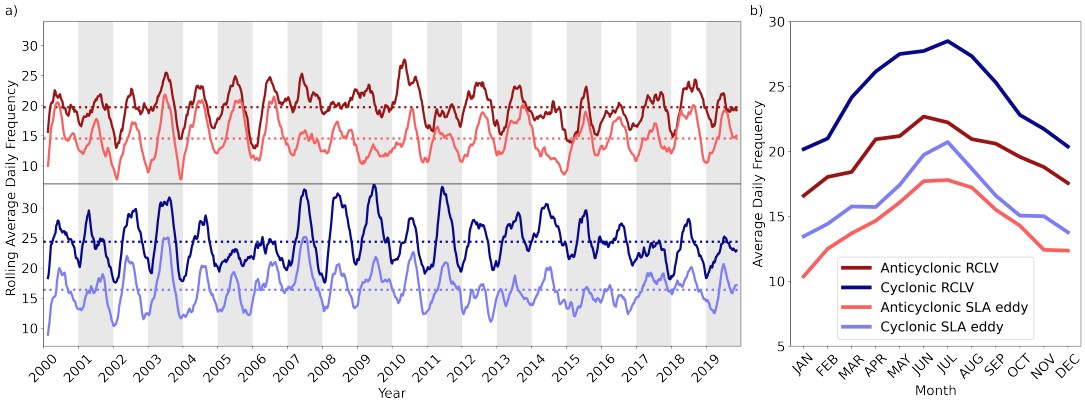

**Figure 6.** (a) RCLV and SLA eddy rolling mean frequency from 2000-2019. Frequency refers to the number of eddies per 8-day timestep. The horizontal dotted lines show the mean frequency for each feature type. (b) Monthly average RCLV and SLA eddy frequency.

32 days of age or older, the seasonal peak presented here may be shifted later in the year than if we were to include the younger eddy phases. We found that, on any given day, there tend to be 25% more anticyclonic and 32% more cyclonic RCLVs than SLA eddies.

Possible explanations for the annual cycle of mesoscale activity include seasonal changes in thermal interactions with the atmosphere (Rieck et al., 2015), seasonality in the conversion of eddy potential energy into kinetic energy (Guo et al., 2022), and the inverse energy cascade of increased submesoscale activity in the winter to mesoscale activity in the summer (Sasaki et al., 2017; Uchida et al., 2017; Schubert et al., 2020). We found no significant trend in changes in eddy frequency from 2000-2019, and deviations from the annual mean often co-occur in the RCLV and SLA eddy datasets (Appendix Fig. C1). Interannual changes are likely driven by large-scale climate variability. For example, submesoscale activity is more active in El Niño winters in this domain, and by the inverse energy cascade, this modulates mesoscale interannual variability (Sasaki et al., 2022). During positive phases of the Pacific Decadal Oscillation, the vertical shear between the Hawaiian Lee Countercurrent and the North Equatorial Current increases, enhancing baroclinic instability and EKE in the Hawaiian Lee Eddy region (Yoshida et al., 2011). Similarly, positive phases of the Pacific Decadal Oscillation drive baroclinic instabilities and increased EKE to the northeast of the Hawaiian Islands (Chen and Qiu, 2010).

### 3.2   Eddy size

We found variability in the eddy area as RCLVs evolve through time, although not as variable as the areas of SLA eddies. For RCLVs that lived longer than the minimum 32 days (N = 7059), the median relative change in areas between the 8-day timesteps is 0.21 and the mean is 0.51 (Appendix Fig. D1). This supports the need for a high temporal resolution to capture details of the boundary evolution of material coherence. A dominant pattern emerges in the changes in area throughout RCLV

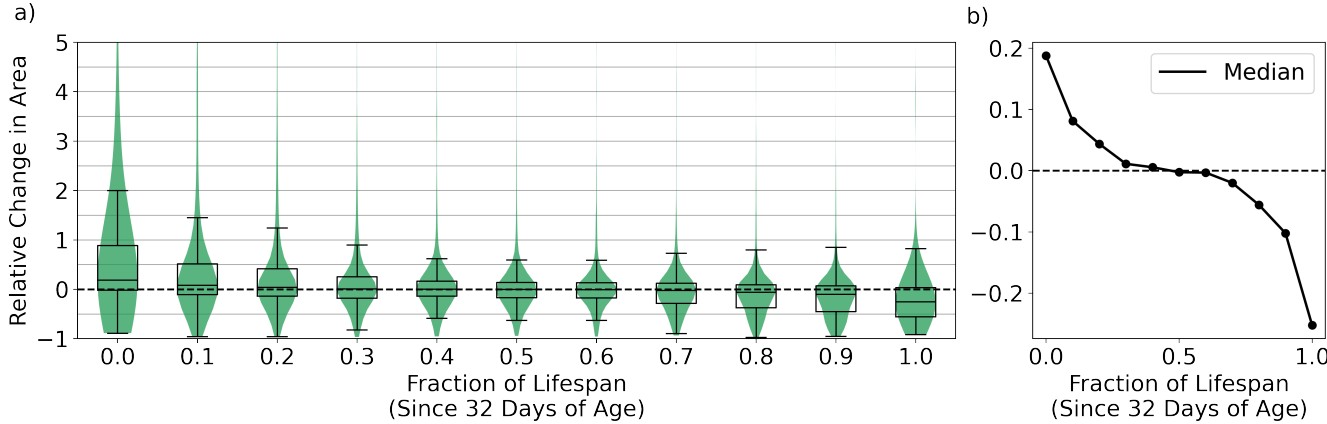

**Figure 7.** (a) Relative change in the RCLV area between 8-day timesteps as a function of the fractional lifespan since the "birth" of the coherent structure at age 32. This only includes RCLVs that live at least 48 days (N = 4962), or three timesteps. The violin plots show the probability distributions in green and the corresponding box plots are overlaid in black. For the box plots, the middle line represents the median, the box spreads the interquartile range, and the whiskers extend to 1.5 times the interquartile range. (b) The median change in area as a function of the fractional lifespan.

lifetimes. We found that following the initial 32-day detection, RCLVs tend to grow for the next third of their lifespan, then maintain their area, and finally decrease in size during the last third of their life (Fig. 7).

SLA eddies tend to be substantially larger than RCLVs, consistent with previous studies (Abernathey and Haller, 2018; Liu et al., 2019; Xia et al., 2022b; Liu and Abernathey, 2023). The RCLV median area is $5,203\ km^2$ and $19,693\ km^2$ for SLA eddies. For the cases of overlapping eddy boundaries, SLA eddies are on average 3.57 times the size of RCLVs (Appendix Fig. D2). As a result, for 33.3% of overlapping eddies, the RCLV is completely contained within an SLA eddy boundary. SLA boundaries are completely contained within an RCLV for 0.91% of the overlapping eddies cases. For the remaining overlapping eddies, the RCLV still tends to be mostly nested within the SLA, with an average of 0.22 of its relative area outside of the SLA eddy bounds (Appendix Fig. D3).

A comparison between eddy maximum area and lifespan is shown in Fig. 8, revealing a positive linear relationship ($R^2 = 0.668$) for RCLVs. In other words, the larger an RCLV, the longer it tends to remain coherent. SLA eddies show a weaker relationship between maximum area and lifespan ($R^2 = 0.405$). For example, there are many cases of large SLA eddies with short lifespans and small SLA eddies with long lifespans. We also tested the relationship between the size and lifespan of the SLA eddies within a similar size range as the RCLVs (maximum area $<= 40,000\ km^2$). Here we found a weaker relationship ($R^2 = 0.377$) between maximum area and lifespan compared to no size restriction (Appendix Fig. D4). These results suggest that RCLV size is more clearly related to eddy longevity than SLA eddies.

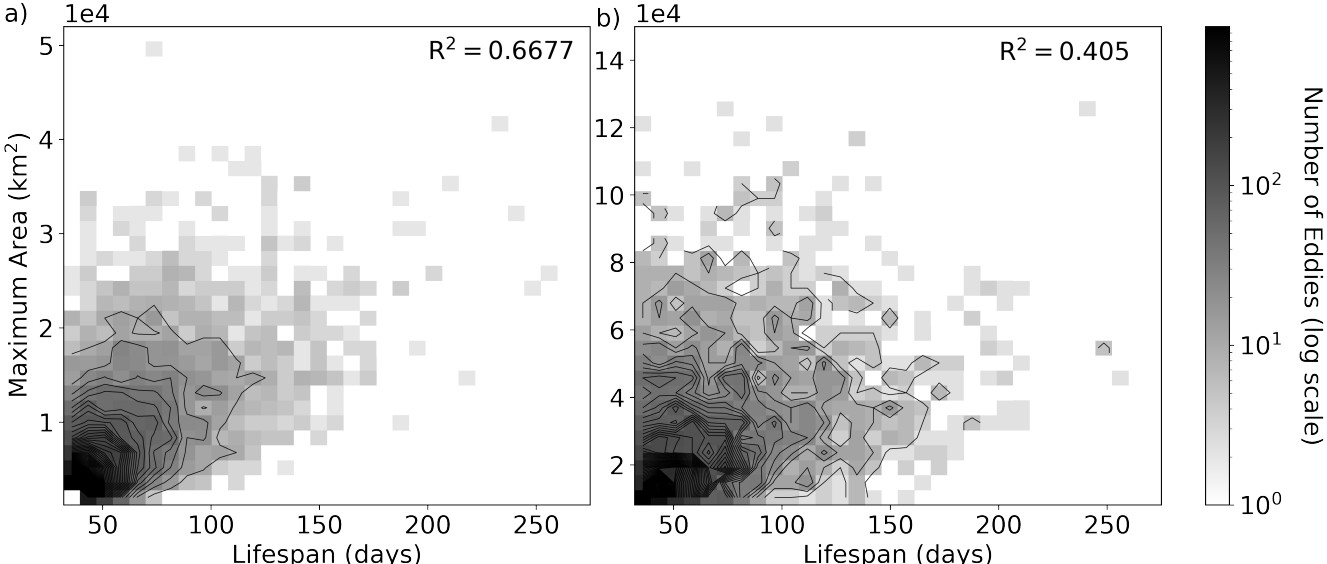

**Figure 8.** 2D histogram of maximum area versus lifespan for (a) RCLV and (b) SLA eddies from 2000-2019. The contours of the histogram distributions are overlaid in black.

## 3.3 Spatial variability

We assessed the spatial variability of the mesoscale activity from the RCLV and SLA eddy atlases across the NPSG. In Fig. 9, eddy frequency was calculated from the number of times each coordinate point was contained within an eddy boundary. The highest frequencies of both RCLVs and SLA eddies are in the Lee of the Hawaiian Islands. Eddies there are sustained by the background currents (Calil et al., 2008; Yoshida et al., 2010; Liu et al., 2012) and the wind stress curl generated where mountains interfere with the northeasterly trade winds (Lumpkin, 1998; Yoshida et al., 2010). The Lee Eddies have two distinct pathways during their westward transit: 1) hugging the islands traveling northwest and 2) southwest. This results in a sideways V-shaped pattern in the spatial eddy frequency to the west of the Hawaiian Islands. Excluding the Lee of Hawaiʻi, eddies are generally evenly distributed across the domain, with marginally more activity to the east of the islands than to the west. The lower frequencies on the domain edges are an artifact of the requirement for the full eddy boundaries to be located inside of the domain. Despite the elevated frequency of RCLVs, we found that the likelihood of being inside an SLA eddy is higher than being in an RCLV because SLA eddies tend to be larger.

The polarity probability (P) is defined

$$P = \frac{(F_A - F_C)}{(F_A + F_C)} \tag{5}$$

where $F_A$ is the frequency of anticyclones, and $F_C$ is the frequency of cyclones Chaigneau et al. (2009). When $P > 0$ ($P < 0$), anticyclonic (cyclonic) eddy polarity is more common at the given location. Figure 10 shows the geographic distribution of

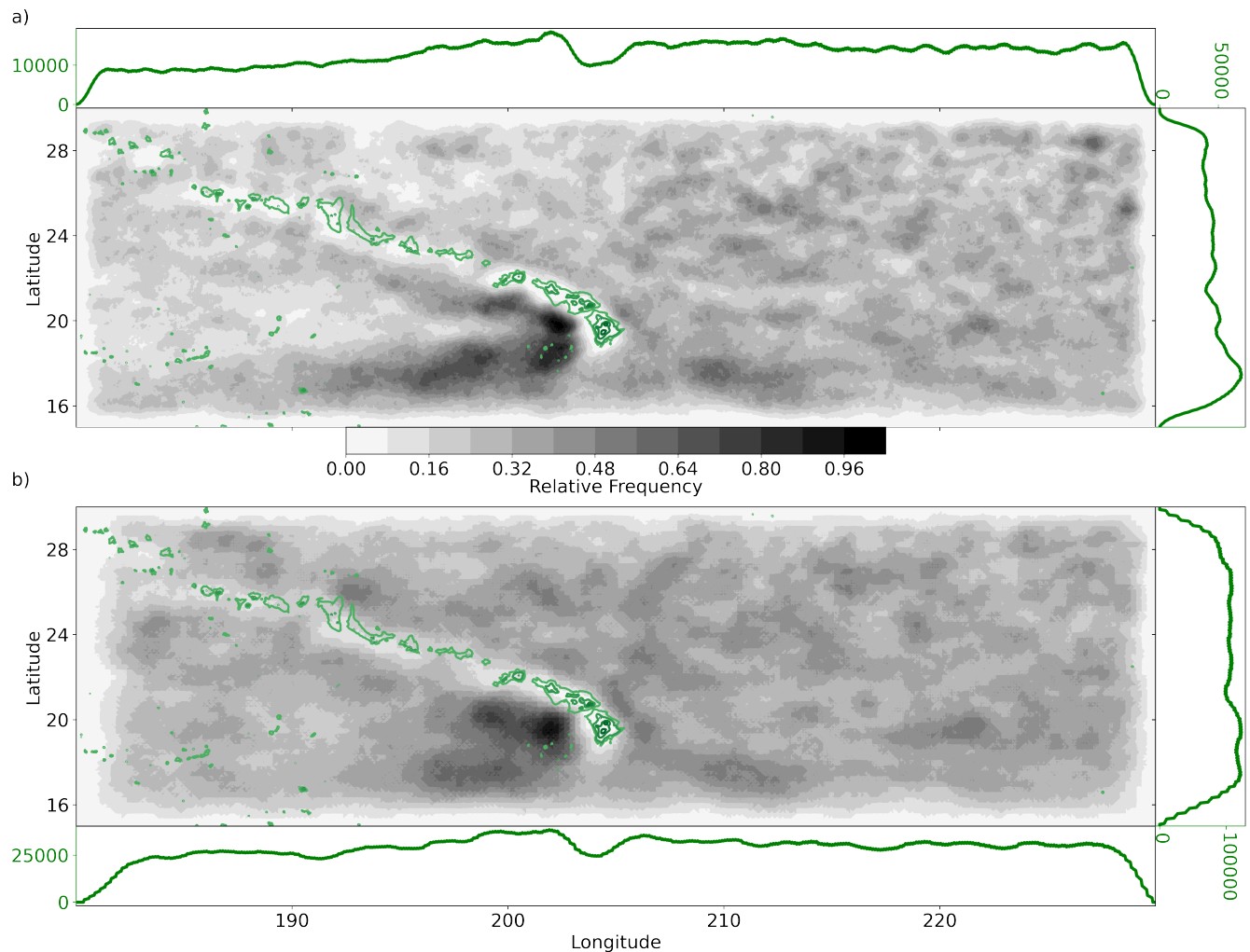

**Figure 9.** Geographic distribution from 2000-2019 of (a) RCLVs and (b) SLA eddies. The green line plots along the horizontal (vertical) represent the number of times an eddy was present at that longitude (latitude). Frequencies include all coordinate points contained within eddy boundaries. The overlaid green contours show the bathymetry (GEBCO, 2023) above 2000 m below sea level, including the Hawaiian Islands and underwater seamounts.

polarity probability of (a) RCLVs and (b) SLA eddies. The polarity probability is similar between the datasets across the
370    domain except to the west of the Islands, in the region of the Hawaiian Lee Eddies.

RCLVs reveal two stark Lee Eddy pathways depending on the polarity, whereas SLA Lee Eddies have a more diffuse polarity probability. This is primarily driven by a disparity in anticyclonic eddy detection between the RCLV and SLA eddy datasets (Appendix Fig. E1). Anticyclonic SLA eddies hugging the west of the islands were detected without any corresponding equivalent in the RCLV atlas. This suggests that anticyclones that propagate close to the Islands are rarely coherent. Between

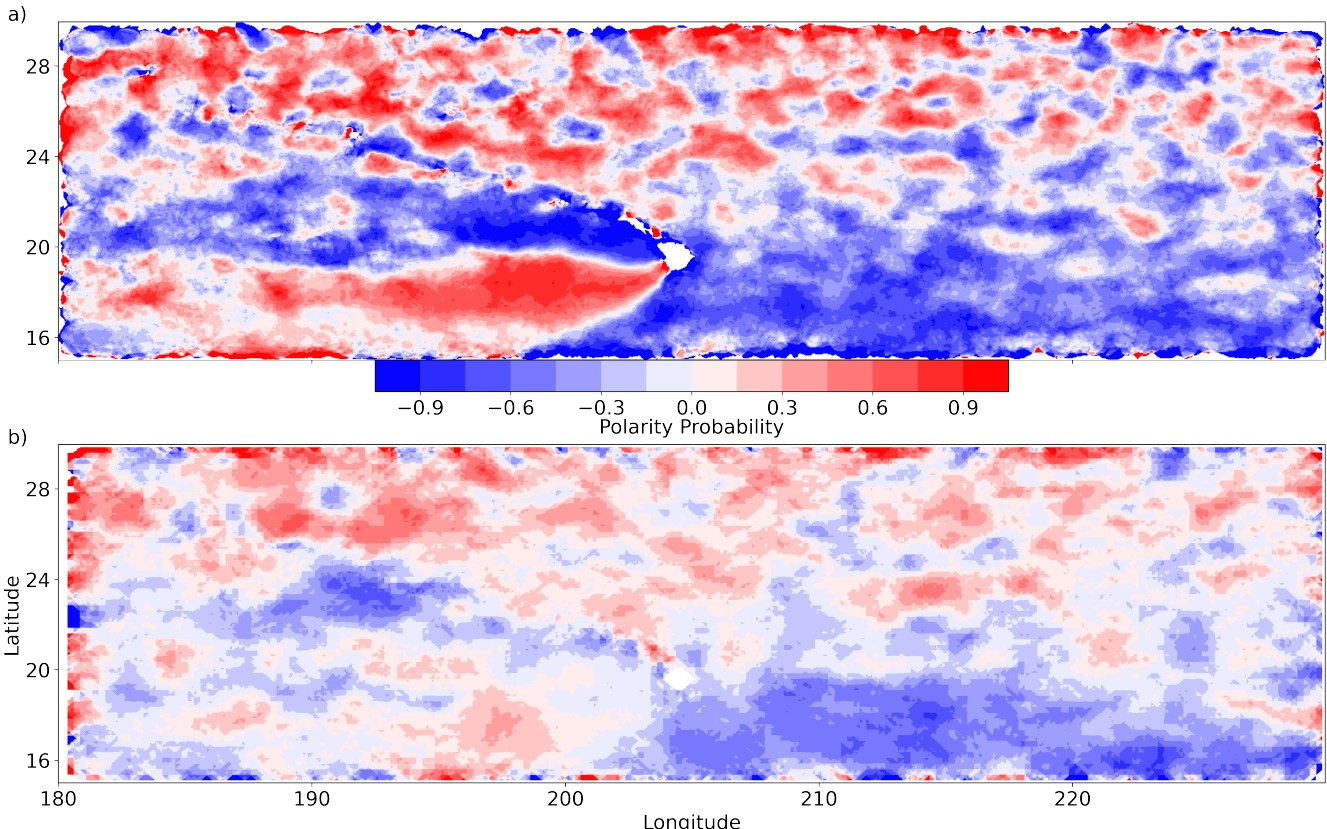

**Figure 10.** Polarity probability from 2000-2019 for (a) RCLVs and (b) SLA eddies. $P < 0$ (blue) indicates more cyclonic activity, and $P > 0$ (red) more anticyclonic activity.

the SLA and RCLV atlases, there is a difference in the location of the most frequent anticyclones that may represent the formation site of the anticyclonic Lee Eddies. The LAVD method likely outperforms SLA-based eddy identification methods in regions with semi-regular or permanent eddies because the baseline sea level is close to the eddy-driven sea level. Cyclonic SLA eddies also occur in the Lee Eddy region dominated by anticyclonic RCLVs, with little to no cyclonic RCLVs in the same location (Appendix Fig. E2). This may be driven in part by large SLA bounds extending further south than the corresponding RCLVs. Cyclones are unable to maintain coherency in the southern Lee Eddy region likely due to the background vorticity generated by the large-scale currents.

There is more anticyclonic activity in the northern half of the domain from 23-30°N in both datasets, whereas cyclones are much more frequent in the southeast quadrant from 15-23°N. This is consistent with previously published studies that find latitudinal bands of eddy polarity probability across the global ocean, including similar structures in our domain (Chelton et al., 2011b; Dong et al., 2022). The process by which these bands are generated remains unclear. Nevertheless, we find distinct regional mesoscale provinces generated by the polarity of eddies, which is of particular interest for biogeochemical

applications since phytoplankton respond differently to cyclones and anticyclones. The RCLV dataset reveals a more striking banded pattern by latitude, whereas in the SLA dataset, the transitions are fuzzier. This divergence is driven by the larger areas of SLA eddies, which act to extend the ranges of eddies of each polarity. To bolster this claim, Appendix Fig. E3 reveals the distributions of polarity exclusively from eddy centers. The differences between the atlases, in this case, are minimal except for in the Lee of Hawai'i (Appendix Fig. E4).

### 3.4 Chlorophyll-*a* in a coherent eddy: A case study

Hawaiian Lee cyclones have been influential in the study and thinking of the mesoscale structuring of chl-*a* and ocean productivity. The cyclonic vorticity is associated with an adiabatic uplift of nutrient-rich isopycnals into the euphotic zone, stimulating primary production and blooms of phytoplankton, a process termed "eddy-pumping" (Falkowski et al., 1991). The resulting chl-*a* blooms in Lee cyclones are documented in both in situ data and remote sensing (e.g., Falkowski et al., 1991; Seki et al., 2001; Vaillancourt et al., 2003; Benitez-Nelson et al., 2007; Benitez-Nelson and McGillicuddy Jr, 2008; Calil and Richards, 2010). The intensity of a mesoscale chl-*a* feature should depend, in part, upon the rate of dilution with surrounding waters. For example, SLA eddies that are continuously refreshed with surrounding waters may have altered mesoscale-attributed biological anomalies compared to a community isolated within the boundaries of an RCLV.

To illustrate the utility of the RCLV atlas, we examined how Lagrangian coherency influences the interpretation of chl-*a* in the long-lived Hawaiian Lee cyclone presented also in Fig. 5 (RCLV ID 6724). Panels (a) and (b) of Fig. 11 show the 8-day average chl-*a* concentration from 9-17 May 2010. On May 17th, the SLA eddy contour was 78 days old and the RCLV eddy contour was 56 days old. This age discrepancy occurred because the eddy was not coherent during early formation, and thus it appears as a dispersive SLA eddy before developing an RCLV in its core. Toward the end of the eddy life, the RCLV persisted for over a month longer than the SLA eddy. An SLA eddy may disappear prematurely, such as in this case, if the SLA anomaly flattens below the detection limit or if the boundary area reduces below the size thresholds. Video S2 shows the eddy contours and chl-*a* evolution from genesis to dissolution.

We expect the coherent core, identified by the RCLV boundaries, to be less affected by lateral dilution with surrounding waters than the outer ring of the SLA feature as demonstrated in Fig. 5. Fig. 11(c) shows the eddy boundaries on May 17th (in black) and April 23rd (in gray). The green line is the trajectory of a Lagrangian particle between those dates. It continues to recirculate inside of the eddy, nested in the interior of the RCLV. The orange line shows the trajectory of another Lagrangian particle that is inside of the SLA bounds, but outside of the RCLV. The orange particle was not inside of the eddy on April 23rd, rather it originated from waters near the Hawaiian Islands. The RCLV continues to recirculate the same water mass internally while the area outside of the RCLV experiences significant exchange with surrounding waters.

The chl-*a* anomalies inside of the respective eddy contours over the feature lifetime are shown in Fig. 11(d). The chl-*a* anomaly is defined as the difference between the mean chl-*a* in the region of twice the radius of the eddy boundary (excluding the eddy) and the mean chl-*a* inside of the boundary. Although the RCLV was nested within the SLA eddy bounds, the chl-*a* anomaly of the SLA eddy exceeded the RCLV anomaly only at one timestep, on July 12th. In this case, the SLA eddy acquired high chl-*a* waters from a large-scale bloom to the northeast (Video S2). To highlight the contribution of the RCLV to the chl-*a*

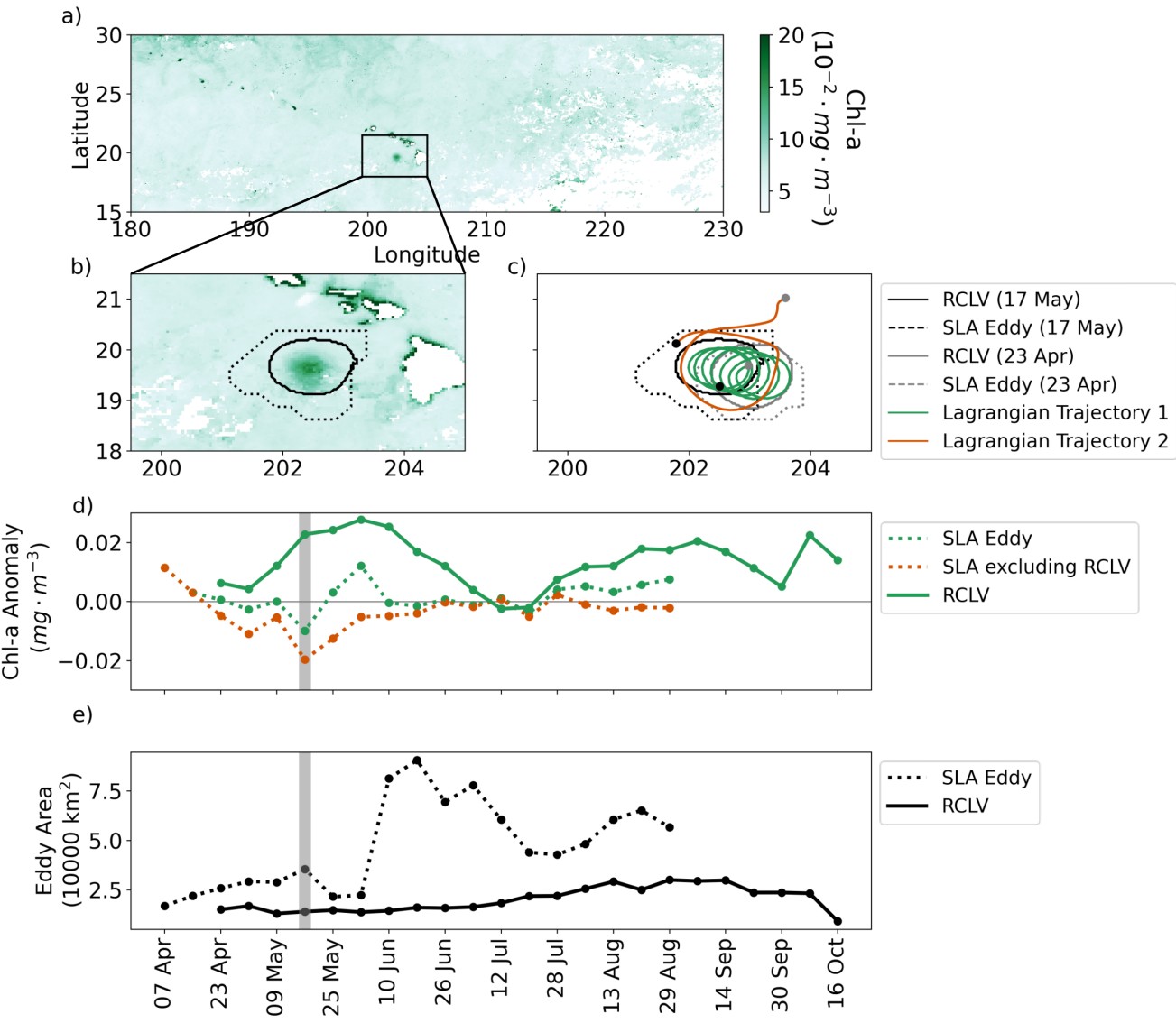

**Figure 11.** Chl-*a* in a Hawaiian Lee cyclone. (a) 8-day average chl-*a* from 9-17 May 2010. (b) Same as (a), but zoomed in on the anomalous chl-*a* patch with the eddy boundaries overlaid. (c) A comparison of eddy boundaries on May 17th and 24 days prior on April 23rd. The Lagrangian trajectories of a particle from the interior of the RCLV (Lagrangian Trajectory 1) and a particle inside of the SLA eddy boundary on May 17th (Lagrangian Trajectory 2) are plotted. (d) Average chl-*a* anomaly inside of the boundaries over the eddy lifetime. The value of the anomaly in the SLA excluding RCLV is equivalent to the SLA anomaly in April before the RCLV was detected. The dots depict the 8-day frequency of the eddy atlases and the gray bar highlights the chl-*a* concentration corresponding with panels (a) and (b) on May 17th. (e) Areas of the SLA eddy and RCLV over the feature lifetime.

anomaly in the SLA eddy, we plotted the anomaly inside of the SLA eddy contour excluding the RCLV (dashed orange line in Fig. 11(d)). In this dispersive zone of the eddy, the chl-*a* anomaly was lower than the all-encompassing SLA eddy at every timestep. A viable interpretation is that the RCLV was trapping and transporting a phytoplankton community, whereas the biomass in the non-retentive zones of the SLA eddy was quickly diluted by mixing with surrounding waters.

Likely due to a shoaling mixed layer, chl-*a* increased in both the SLA and RCLV in May, the month that the mixed layer depth is the shallowest in the region (Rii et al., 2022). In June, the RCLV maintained a relatively constant area, whereas the SLA eddy dramatically increased in size (Fig. 11(e)). During this change in area, the chl-*a* anomaly was elevated in the RCLV but quickly driven to 0 in the SLA eddy, suggesting that concentrations were diluted via exchange with surrounding waters outside of the coherent structure. The magnitude of the positive chl-*a* declined in the RCLV for several weeks until July when

concentrations were lower than the large-scale bloom in surrounding waters. Beginning in August, chl-*a* was anomalously high again in the eddy, when there was a bloom isolated inside the coherent structure, compared to low chl-*a* in surrounding waters. The end of August marked the demise of the SLA feature, but the chl-*a* concentration continued to be higher inside of the RCLV than in surrounding waters. This highlights that coherent mesoscale circulations may continue to modulate chl-*a* beyond the detection limit of the SLA method. A large-scale bloom developed in September (Video S2), the signatures of

which were enhanced inside of the coherent structure, over a month past the detection of the SLA eddy.

## 4    Discussion

We presented an atlas of Lagrangian coherent mesoscale eddy boundaries in the North Pacific Subtropical Gyre (NPSG), spanning two decades and designed with several unique features focused toward biogeochemical applications. Key aspects of the methodology are the 8-day frequency synchronized to ocean color products, backward-in-time Lagrangian analysis, and

eddy tracking. By comparing Lagrangian (RCLVs) and Eulerian (SLA) mesoscale eddy boundaries, we showed that there are clear distinctions in size and lifespan. On the other hand, the temporal and spatial variabilities are similar for RCLVs and SLA eddies, suggesting that both datasets capture mesoscale activity that is modulated by analogous mechanisms.

We tracked 11,900 unique RCLVs and roughly half as many SLA eddies. Only half of the eddies detected from the SLA contain a Lagrangian coherent structure (as defined in this study). However, two-thirds of SLA eddies overlap with RCLV

at some point in their lifetime. In other words, a third of mesoscale eddies detected from the SLA are never coherent and experience a significant exchange of material with surrounding waters over their whole lifetime. Only 36% of the tracked RCLVs are concurrent with an SLA eddy boundary at some point in their lifetime, indicating that a majority of coherent eddies may be missed by SLA identification entirely.

RCLVs that do not overlap with an SLA eddy for their entire lifetime have significantly shorter lifespans (median 32 days)

than their overlapping counterparts (median 72 days). Of the tracked RCLVs that do overlap with an SLA during their lifetime, 54% develop prior to the detection of the SLA eddy boundary and 66% continue to propagate after the concurrent SLA eddy is no longer detectable. This suggests that the majority of the occurrences of non-overlapping RCLVs are short-lived features,

developing SLA eddies, or the remnants of SLA eddies. We hypothesize that elevated chl-*a* at the onset of SLA eddy detection may be enhanced due to early trapping of an initial anomaly.

Using the definitions applied here, RCLVs tend to be much smaller than SLA eddies. We found that the areas of the RCLV boundaries have a positive linear relationship with lifespan (Fig. 8). The comparatively large SLA boundaries have a weaker relationship with eddy lifespan and act to dilute the underlying spatial distribution of eddy polarity as manifested in the RCLV dataset (Fig. 10). Although RCLV and SLA eddy boundaries encompass mechanistically similar or equivalent mesoscale features, it is important to consider the lenient boundaries derived from SLA detection methods when interpreting biogeochemical

observations categorized as "inside" or "outside" of an eddy.

  We detected many coherent vortices that have an elongated shape. Obvious examples of this include the overlapping anticyclonic RCLV and SLA eddy north of the islands and below the legend at coordinates $(202.3°, 22.8°N)$ in Fig. 1, and the RCLV on July 4th in Fig. 3. Commonly used eddy detection algorithms (e.g., AVISO products Chelton et al., 2011b; Mason et al., 2014) penalize features that are less circular and potentially miss coherent mesoscale eddies. An advantage of the OceanEddies

algorithm used here is that it is parameter-free with respect to eddy shape (Faghmous et al., 2015). Nevertheless, it would be informative to directly compare the RCLV atlas with other eddy datasets. We included a comparison of the RCLV and SLA eddy datasets to the OW parameter from a single day in Appendix Fig. A1.

### 4.1 Hawaiian Lee Eddies and implications for chlorophyll-*a*

The Hawaiian Lee Eddies generate the strongest signals of eddy activity in this domain, falling into two distinct groups. The

first are long-lived anticyclonic eddies generated by the instability of the shear between the eastward flowing Hawaiian Lee Countercurrent and the westward flowing North Equatorial Current (Calil et al., 2008; Yoshida et al., 2010; Liu et al., 2012). These anticyclones travel westward in the latitude band between 15-20°N. The second group includes cyclones induced by a wind stress curl anomaly associated with the blocking of the northeasterly trade wind (Lumpkin, 1998; Yoshida et al., 2010). These cyclones hug the western side of the Hawaiian Islands while transiting toward the Hawaiian Lee Current between 20-

24°N (Lumpkin, 1998). We found that both groups of Lee Eddies have predictable translation pathways, are strongly coherent, and are long-lived. However, anticyclonic Lee Eddies that hug the islands only appear in the SLA eddy dataset. This suggests that these Lee Eddies are not coherent on the timescales explored in this study.

  The consistent generation and pathways of Hawaiian Lee cyclones have made them a key tool in the study of the biogeochemistry of ocean eddies (e.g., Falkowski et al., 1991; Seki et al., 2001; Vaillancourt et al., 2003; Benitez-Nelson et al., 2007;

Benitez-Nelson and McGillicuddy Jr, 2008; Calil and Richards, 2010). However, due to their strong trapping capacity and consistent paths of propagation, they may not be representative of the shorter-lived and less coherent eddies generated elsewhere in the domain (e.g., at Station ALOHA located north of the Hawaiian Islands (Barone et al., 2019; Rii et al., 2022)). We hypothesize that the biological signature of eddies will vary depending on their trapping behaviors and how eddy boundaries are defined, as exhibited in Fig. 11. We have a follow-up manuscript in preparation using the RCLV atlas presented here to

examine the influence of eddy trapping on surface chl-*a* across the domain.

## 4.2 Even RCLVs are "leaky"

A consideration when employing any such analysis with the geostrophic approximation is that the identified coherent features are not truly closed systems at the ocean surface. The ageostrophic component of the flow is not resolved in the SLA-derived flow fields used to evaluate particle trajectories in this study. In fact, the LAVD method fails to identify RCLVs at the surface in high-resolution simulated velocity fields that include ageostrophic motions (Sinha et al., 2019; Xia et al., 2022b; Ntaganou et al., 2023). There can be mixing at the margins of eddies due to submesoscale instabilities, and the two-dimensional nature of this analysis misses the effects of vertical motions and mixed layer thickening. Despite these considerations, we still observed concentrated patches of chl-*a* within coherent eddy boundaries, such as in Fig. 11. This suggests that, in this case, the timescale of unresolved fluxes through the coherent boundaries is slower than the biological processes, and thus the localized bloom is contained or enhanced. Mesoscale geostrophic structures penetrate below the mixed layer in this region (Liu et al., 2012, 2020), so RCLVs identified from the surface geostrophic flow may also reflect lateral trapping deep in the water column where chl-*a* tends to be at a maximum. In a 3D simulation of the Gulf of Mexico and the Caribbean Sea, Ntaganou et al. (2023) identify RCLVs as deep as 800 meters. There is a need for more estimates of RCLV penetration depth and an improved understanding of the influence of the ageostrophic flow on eddy trapping.

## 5 The value of an RCLV atlas for biogeochemical applications

The biogeochemical applications of SLA eddy atlases frequently relate to the evaluation and interpretation of microbial communities, primary productivity, and carbon export. Such interpretation often hinges on the assumption that most SLA eddies are nonlinear features that trap and transport fluid over conceivably large, lateral distances (Chelton et al., 2011b, a). Under the assumption of coherency, the development of biogeochemical properties of SLA eddy interiors could be interpreted in a simple, Lagrangian frame. However, clouding such presumptions, our analysis shows definitively that approximately half of the SLA-identified features in the NPSG are not coherent from a Lagrangian perspective. Furthermore, for the eddies that do contain a coherent structure, SLA-derived boundaries encompass large areas that do not delineate between dispersive and coherent zones. By identifying RCLV boundaries and capturing their evolution, we can exploit natural perturbation experiments with which to interpret system dynamics: the most coherent mesoscale features can appropriately be considered temporarily "closed" Lagrangian systems.

In summary, we presented a publicly available RCLV atlas (2000-2019) characterizing the natural history of Lagrangian coherent eddy boundaries in the NPSG. Taking advantage of the time-synchronization to remotely sensed ocean color products, high temporal resolution, and backward-in-time coherency of this atlas, the impact of mesoscale eddy trapping on the local biogeochemistry can be addressed. The atlas can be employed in retrospective analysis of observational datasets (e.g., Barone et al., 2019; Rii et al., 2022) and numerous biogeochemical expeditions in the region that targeted mesoscale features (e.g., Seki et al., 2001; Vaillancourt et al., 2003; Benitez-Nelson et al., 2007; Benitez-Nelson and McGillicuddy Jr, 2008; Harke et al., 2021; Zhang et al., 2021; Barone et al., 2022). Additionally, the RCLV atlas is a contextual tool that may aid in future

cruise planning and in situ eddy sampling. The applications of this dataset are not limited to biogeochemical applications, as it may also be useful for studies of the dynamical transport of marine plastics, pollution, and heat.

## 6 Code and data availability

This study was conducted using E.U. Copernicus Marine Service Information, namely the $1/4°$ SLA and geostrophic velocity gridded global ocean dataset (Version 008_047), distributed at https://doi.org/10.48670/moi-00148. Daily and 8-day average chl-*a* is produced by OC-CCI (Version 6.0 used here) and distributed by the European Space Agency at https://www.oceancolour.org/ (Sathyendranath et al., 2019). The OceanEddies MATLAB software that detects and tracks Eulerian SLA eddy contours is available at https://github.com/ifrenger/OceanEddies (Faghmous et al., 2015). The OceanParcels v2.0 Python package used to run Lagrangian particle simulations is available at https://oceanparcels.org/index.html (Delandmeter and van Sebille, 2019). The floater Python package available at https://github.com/ocean-transport/floater is used to identify RCLV contours and determine their CD. All figures were created with Matplotlib 3.3.4 (Caswell et al., 2021; Hunter, 2007), available under the Matplotlib license at https://matplotlib.org/. GEBCO 2023 bathymetric data were obtained from https://download.gebco.net/.

The Python software developed for this study is available at https://github.com/lexi-jones/RCLVatlas (Jones-Kellett, 2023b). The code includes a pipeline to run OceanParcels on CMEMS data, a custom kernel to calculate the relative vorticity along particle trajectories, and an RCLV tracking algorithm. The Jupyter notebook script https://github.com/lexi-jones/RCLVatlas/blob/main/example_usage.ipynb includes an example usage code for the software. The NPSG RCLV dataset is publicly available and distributed by Simons CMAP at https://simonscmap.com/catalog/datasets/RCLV_atlas (Jones-Kellett, 2023a).

*Video supplement.* The 2010 eddy field overlaid on the (a) LAVD and (b) SLA.

*Video supplement.* Chl-*a* near the Hawaiian Islands from 22 Mar - 01 Nov 2010, highlighting the evolution of a Hawaiian Lee cyclone. The solid contours represent the RCLV and the dashed represents the SLA eddy.

## Appendix A: Okubo-Weiss Parameter

The Okubo-Weiss (OW) Parameter is a measure of deformation and rotation defined

$$OW = S_n^2 + S_s^2 - \omega^2 \tag{A1}$$

where $S_n$ is the strain normal to the flow, $S_s$ is the shear strain, $\omega$ is the relative vorticity (Okubo, 1970; Weiss, 1991). These are expressed as

$$S_n = \frac{\partial u}{\partial x} - \frac{\partial v}{\partial y}, S_s = \frac{\partial v}{\partial x} + \frac{\partial u}{\partial y}, \omega = \frac{\partial v}{\partial x} - \frac{\partial u}{\partial y} \tag{A2}$$

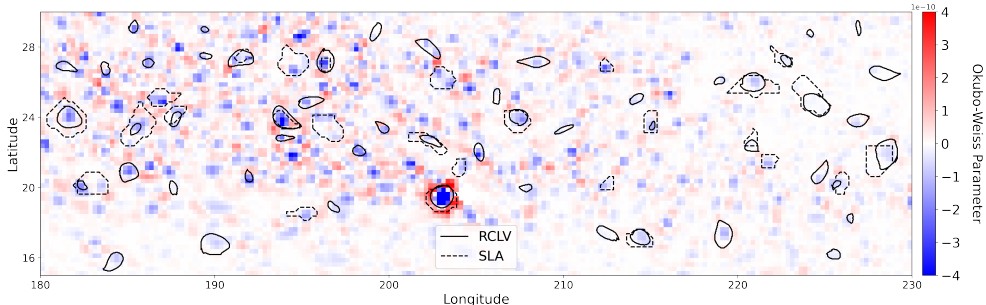

**Figure A1.** The Okubo-Weiss parameter from 2010 April 23 with RCLV contours (solid lines) and SLA eddy boundaries (dashed lines) plotted on top.

The OW parameter is sometimes used to identify eddies from satellite geostrophic velocity by detecting circular regions of the flow that switch from strain-dominated to vorticity-dominated.

In Fig. A1, the blue regions are vorticity-dominated ($OW < 0$) and the red are strain-dominated ($OW > 0$). Every rotationally coherent Lagrangian vortex (RCLV; described in Sections 1.3 and 2.4) in Fig. A1 encompasses a region of negative OW because both metrics are functions of the relative vorticity. On the other hand, not every low-magnitude OW patch corresponds with an RCLV. The OW parameter is an Eulerian measurement, so it does not detect variations in vorticity in features through time. The OW method is observer-dependent and therefore not sufficient as a standalone metric to determine coherent features of the flow (Haller, 2015). Beron-Vera et al. (2019) shows that the material inside of eddy boundaries derived using the OW method can be more dispersive over time than from boundaries derived from the SLA.

## Appendix B: Coherency Index and Convex Deficiency

The result of testing the *floater* algorithm on one day is shown in Appendix Fig. B1. Similar to the results of Tarshish et al. (2018), we find that the median CI decreases as the target CD increases. One might then assume that features with a higher CD tend to be more dispersive, but this is not always the case. There can be a huge spread of CI values from contours identified with any given target CD. For our purposes, a target CD or cutoff is not sufficient as a stand-alone metric to identify coherent features. To address this problem, our algorithm iterates through three potential target CDs $(0.03, 0.02, 0.01)$. Starting with target $CD = 0.03$, we check if the outputted contour has an actual $CD <= 0.03$. In some cases, the actual CD may be much higher than the target if the algorithm cannot find a contour with the desired CD. If the actual CD passes the criterion, then the CI is calculated for the corresponding contour by tracking all interior Lagrangian particles backward in time for 32 days from the same trajectory data used to produce the LAVD fields. If $CI >= -0.5$ and the sign of the vorticity for 85% of the particles is consistent from day 0 to day -32, then the contour is accepted as an RCLV. Otherwise, the algorithm iterates through the remaining target CDs. In some cases, no contours will qualify at any of the target CDs.

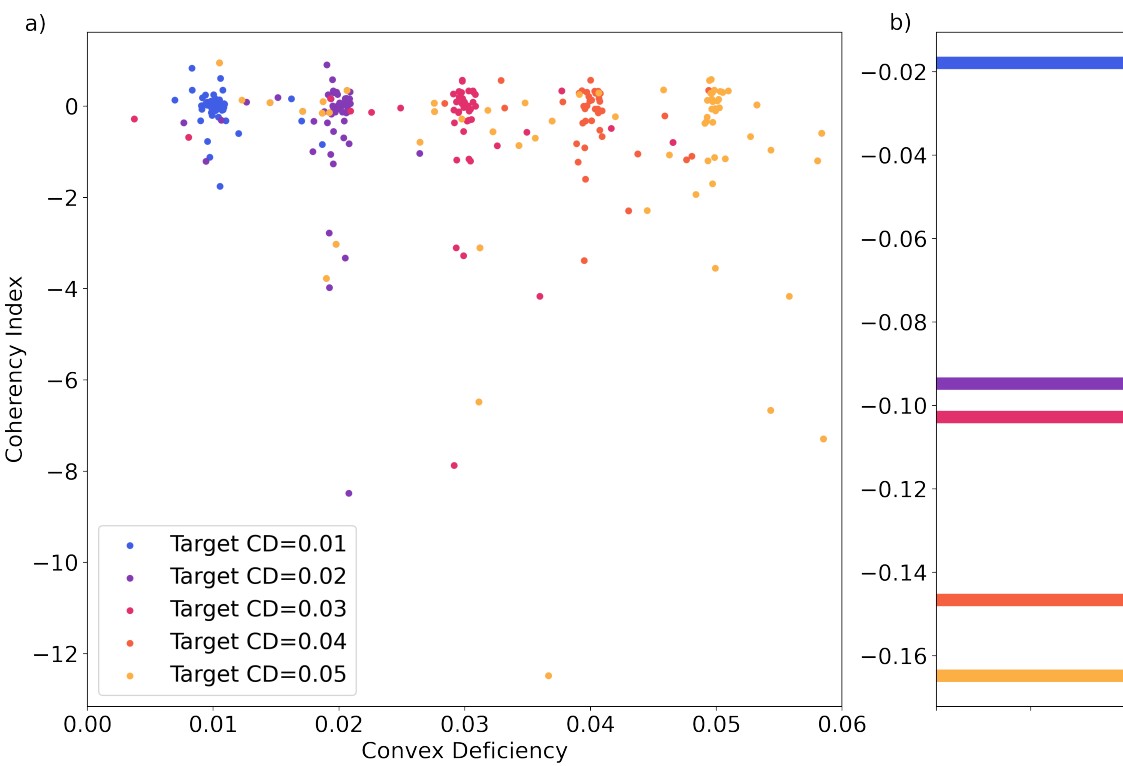

**Figure B1.** Coherency metrics of RCLV contours on 23 April 2010. (a) CI as a function of the actual CD for various target CDs. (b) Median CI for each target CD.

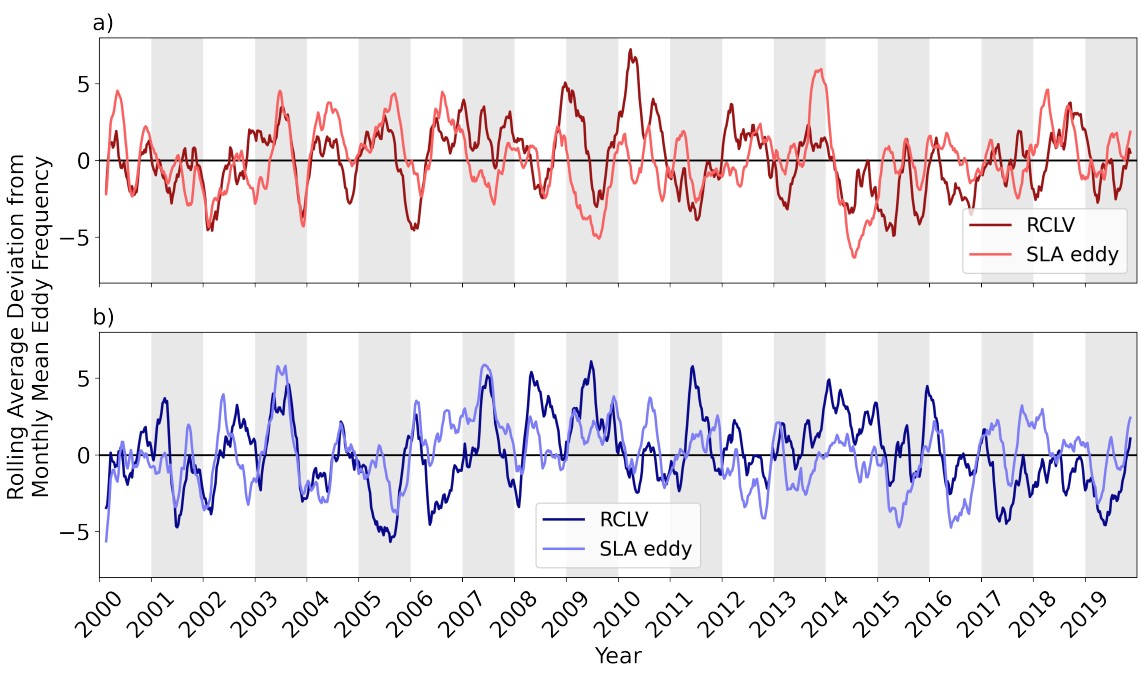

**Figure C1.** Interannual variability of the number of eddies. (a) Rolling mean daily deviation of the anticyclone frequency from the 2000-2019 monthly mean. (b) Same as (a) but for cyclones.

**Table C1.** Monthly mean eddy frequency from 2000-2019

|  | Jan | Feb | Mar | Apr | May | Jun | Jul | Aug | Sep | Oct | Nov | Dec |
|---|---|---|---|---|---|---|---|---|---|---|---|---|
| All RCLVs | 36.76 | 39.04 | 42.59 | 47.09 | 48.69 | 50.43 | 50.74 | 48.29 | 45.91 | 42.40 | 40.53 | 37.94 |
| RCLV Anti | 16.59 | 18.04 | 18.43 | 20.94 | 21.19 | 22.69 | 22.24 | 20.95 | 20.60 | 19.58 | 18.80 | 17.56 |
| RCLV Cyc | 20.18 | 21.00 | 24.16 | 26.15 | 27.51 | 27.74 | 28.50 | 27.34 | 25.31 | 22.82 | 21.73 | 20.34 |
| All SLAs | 23.83 | 26.93 | 29.46 | 30.38 | 33.51 | 37.45 | 38.53 | 35.90 | 32.11 | 29.35 | 27.44 | 26.14 |
| SLA Anti | 10.38 | 12.50 | 13.70 | 14.66 | 16.08 | 17.71 | 17.80 | 17.23 | 15.51 | 14.29 | 12.43 | 12.36 |
| SLA Cyc | 13.46 | 14.43 | 15.76 | 15.72 | 17.43 | 19.74 | 20.73 | 18.68 | 16.60 | 15.06 | 15.01 | 13.78 |

## Appendix C:  Temporal Variability

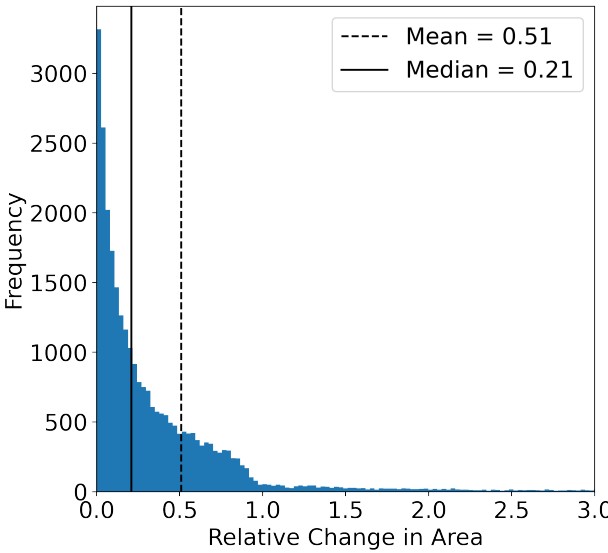

**Figure D1.** Distribution of the absolute value of the relative change in area between 8-day timesteps for RCLVs that live longer than the minimum 32 days (N = 7059).

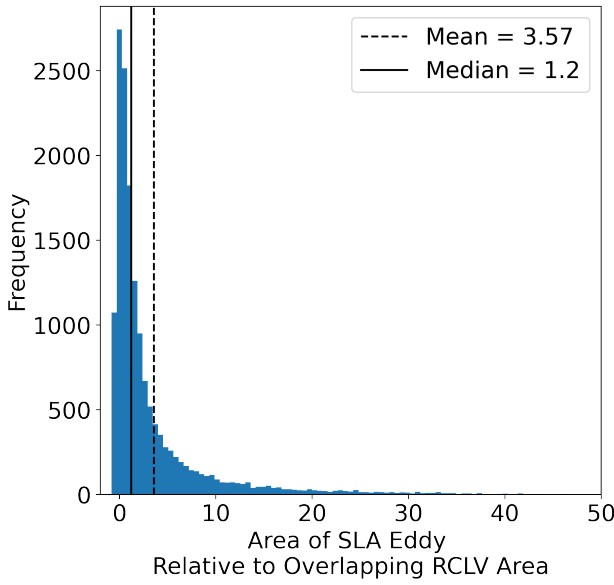

**Figure D2.** Distribution of the relative difference in area between overlapping SLA eddies and RCLVs (N = 15568).

**Appendix D: Eddy area**

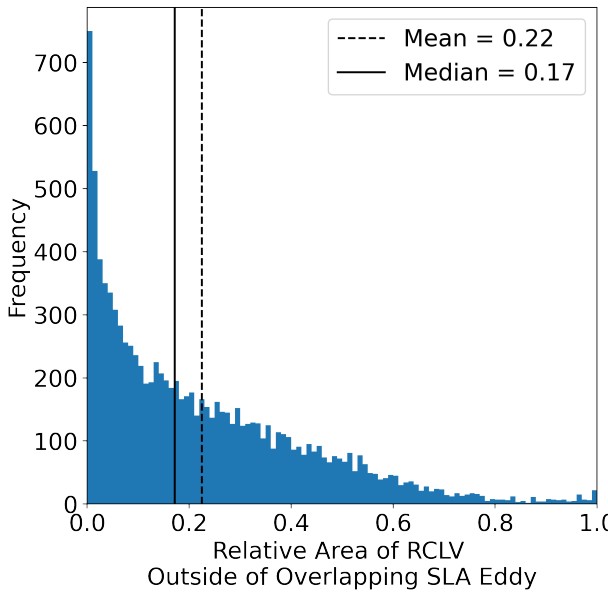

**Figure D3.** Distribution of the relative area of RCLVs outside of their corresponding overlapping SLA eddies. This does not include RCLVs that are completely contained within SLA eddies, or RCLVs that completely contain SLA eddies.

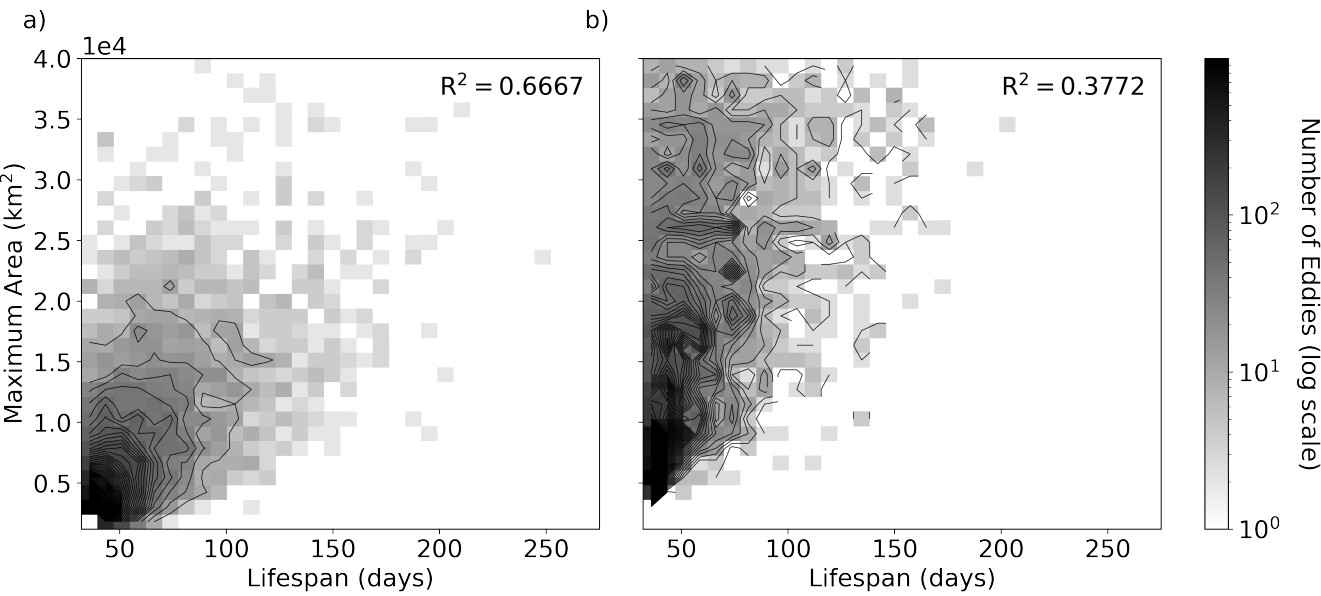

**Figure D4.** 2D histogram of maximum area versus lifespan for (a) RCLV and (b) SLA eddies from 2000-2019 with a maximum area $<= 40,000 \ km^2$. The contours of the histogram distributions are overlaid in black.

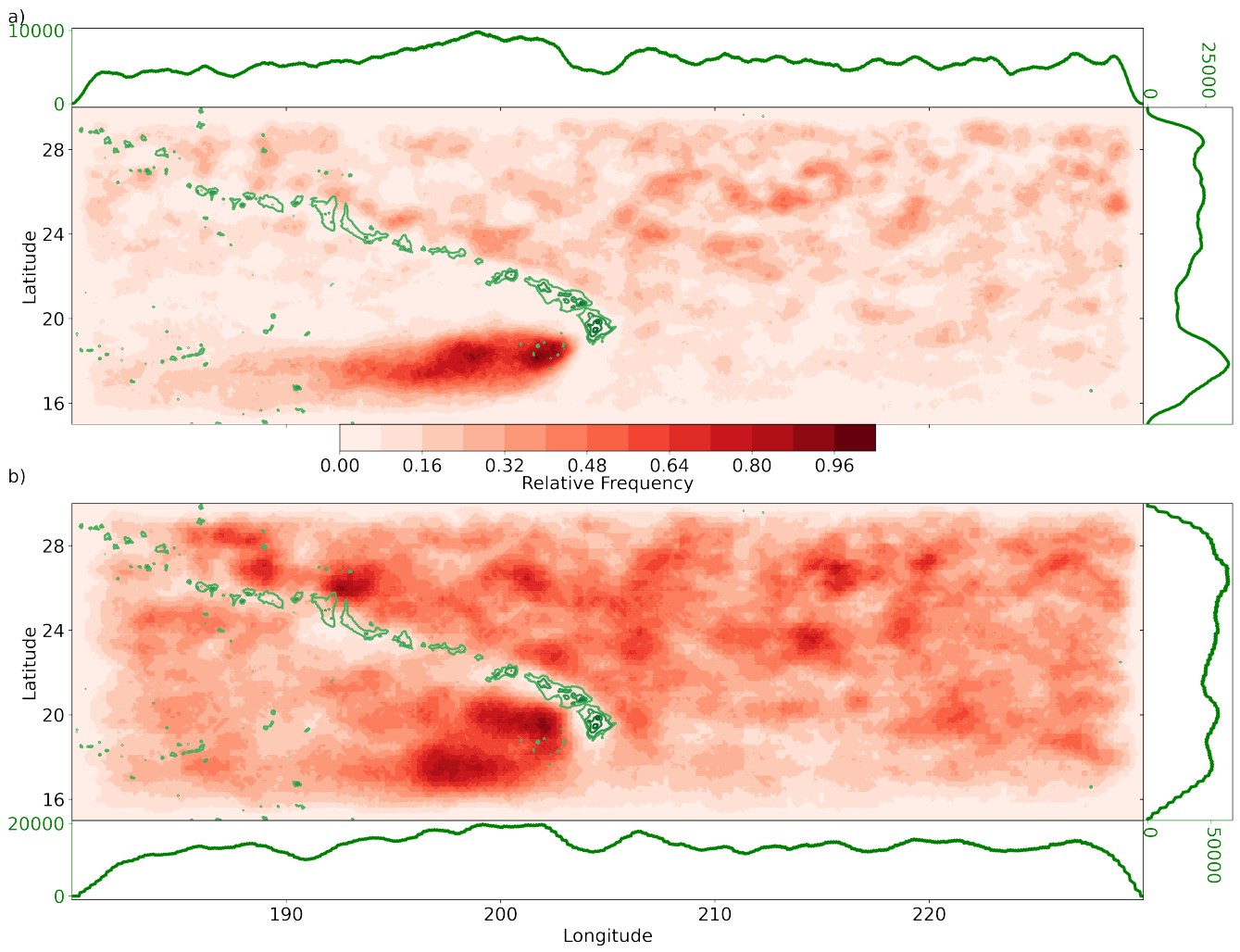

**Figure E1.** Geographic distribution from 2000-2019 of anticyclonic (a) RCLVs and (b) SLA eddies. The green line plots along the horizontal (vertical) represents the number of times an eddy was present at that longitude (latitude). Frequencies include all coordinate points contained within eddy boundaries. The overlaid green contours show the bathymetry (GEBCO, 2023) above 2000 $m$ below sea level, including the Hawaiian islands and underwater seamounts.

## Appendix E:  Spatial distributions

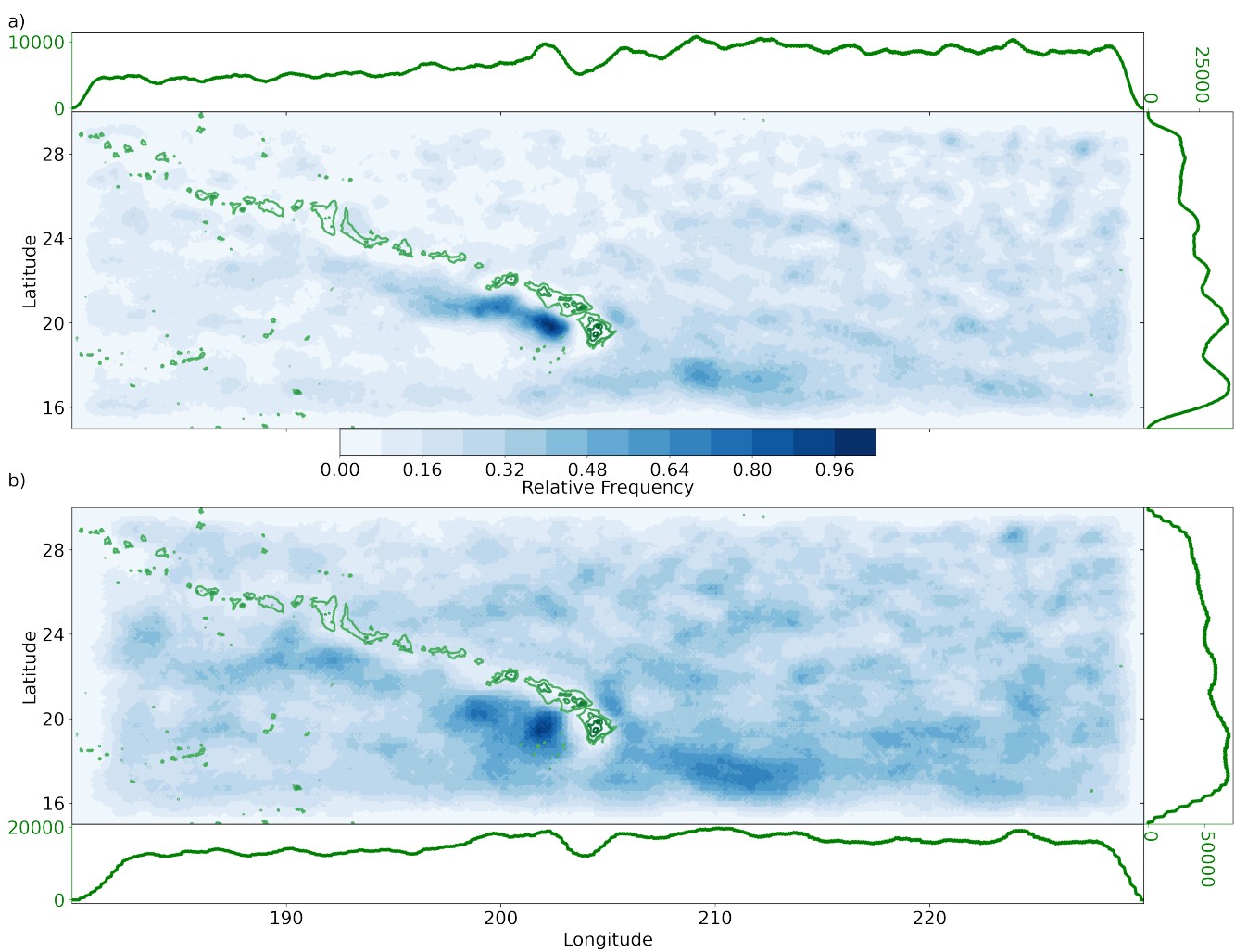

**Figure E2.** Same as Fig. E1 but for cyclonic (a) RCLVs and (b) SLA eddies.

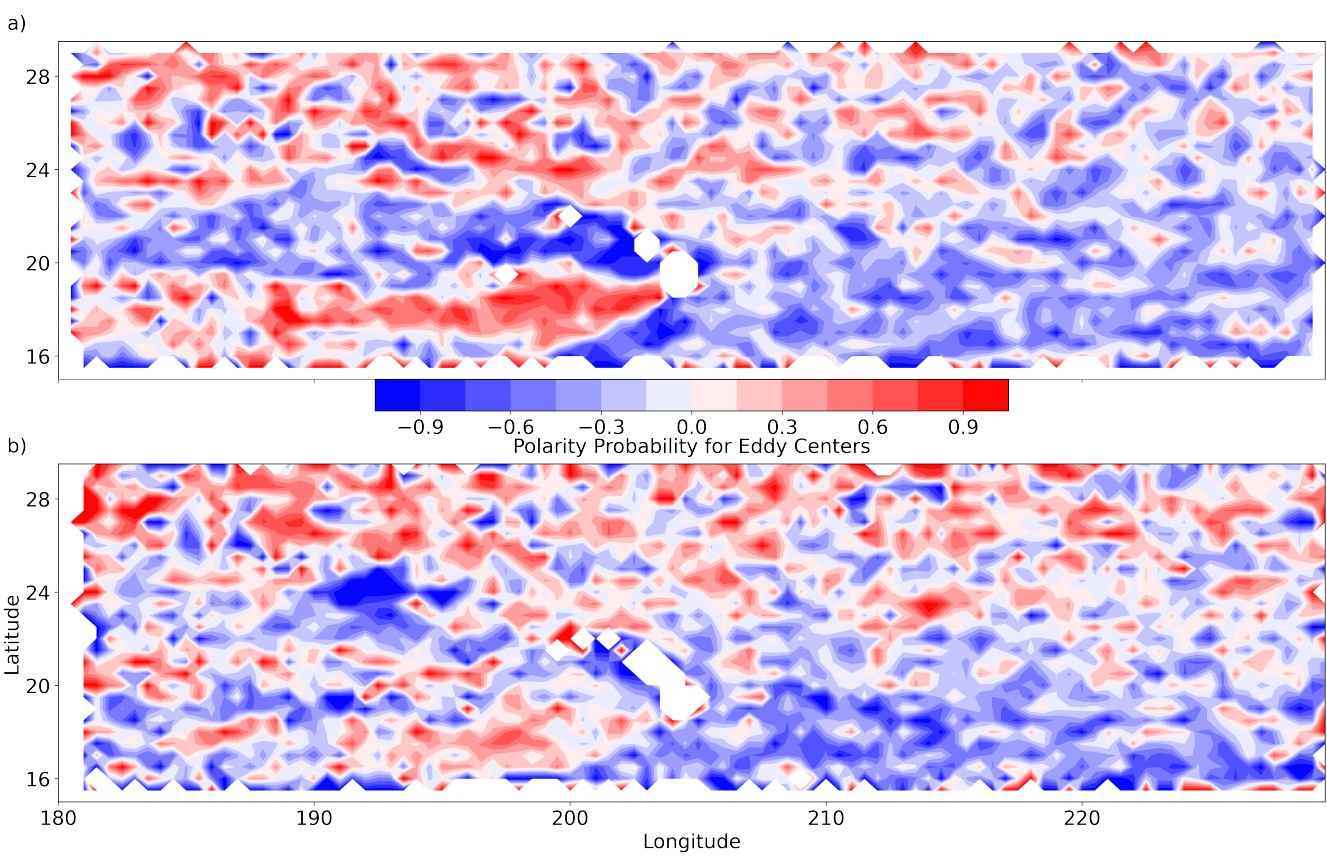

**Figure E3.** Polarity probability of eddy centers from 2000-2019 for (a) RCLVs and (b) SLA eddies. $P < 0$ (blue) indicates more cyclonic activity, and $P > 0$ (red) more anticyclonic activity.

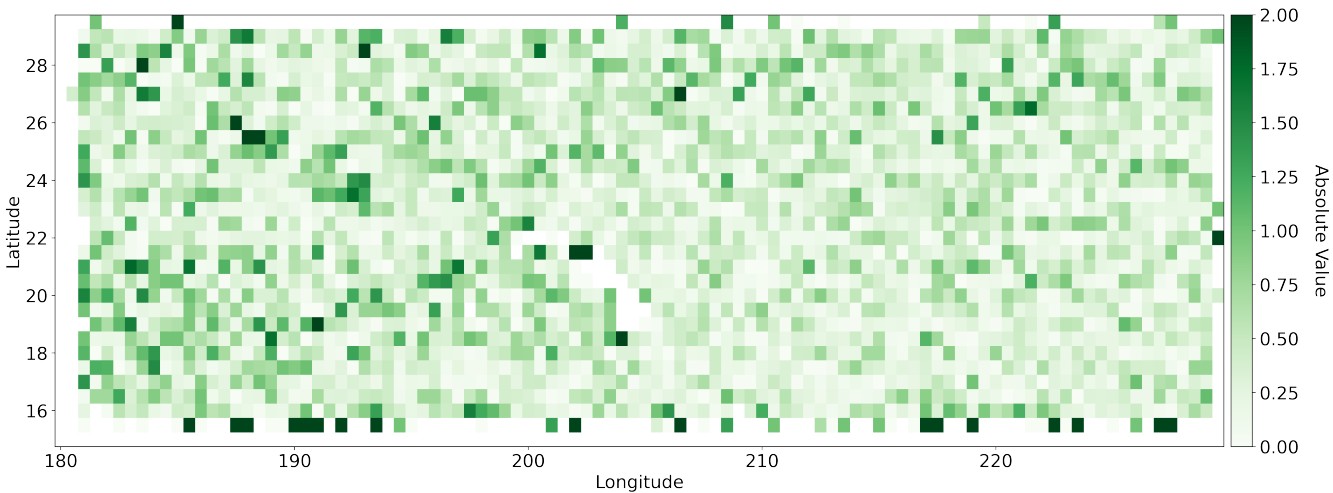

**Figure E4.** Absolute value of the difference in polarity probability of eddy centers between the RCLV (Fig. E3(a) and SLA eddy atlases (Fig. E3(b)).

*Author contributions.* The project conceptualization and methodology were developed by all co-authors. AEJK wrote the software, curated the dataset, produced the figures, and conducted the formal analysis, investigation, and validation. AEJK wrote and prepared the manuscript with significant contributions from MJF. MJF acquired funding and resources for the execution of the project.

*Competing interests.* The authors declare that they have no conflict of interest.

*Acknowledgements.* We thank Stephanie Dutkiewicz, Enrico Ser-Giacomi, Stephanie Anderson, Christopher Hill, Gael Forget, and Danling Ma for their invaluable discussions on Lagrangian methodologies and applications. Stephanie Dutkiewicz, Enrico Ser-Giacomi, and reviewers, including Peter Cornillon and Giuseppe Manzella, provided helpful feedback that significantly improved this manuscript. We are grateful for support from the Simons Foundation (Simons Collaboration on Ocean Processes and Ecology, Award 329108, M.J.F.; CBIOMES, Award 549931, M.J.F.).

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
