# Peer review of "A Lagrangian coherent eddy atlas for biogeochemical applications in the North Pacific Subtropical Gyre"

_Earth System Science Data, 2023_

## Referee Comment (RC2)

[referee-annotated manuscript omitted]

---

## Author Response (AR1)

**Summary of Changes**

We thank Drs. Manzella and Cornillon for their thoughtful and stimulating reviews. Addressing these comments has, we feel, led to a clearer and much-improved revised manuscript. We have addressed each of the reviewers' comments (which are copied directly below in black font) and describe any of our associated changes. To address the comments from RC1 (Dr. Manzella), we substantially expanded the literature review in the Introduction, explicitly highlighted the novel features of the presented RCLV atlas, and clearly described the open questions it was used to address. In response to RC2 (Dr. Cornillon), we provided additional statistics and clarifications in the text, updated three of the original figures (now Figs. 1, 8, 11), and incorporated two new figures (Fig. 5, 7) into the main text and one new figure in the Appendix (D4).
* * *
**RC1: Giuseppe M.R. Manzella**

General Comment

The paper present Lagrangian Coherent Structures as part of a set of methods for identifying coherent eddies and tracking them. The aim is: compute the coherency of the recent past, synchronize the analysis with ocean color products, and provide a high temporally resolved atlas of coherent eddy boundaries. There are many researches on 'eddy' identification with automatic algorithms, machine learning, etc. Examples of papers on this issue are:

> ESSD, 14, 1087–1107, 2022: 1exp: a new global mesoscale eddy trajectory atlas derived from altimetry
> Assessment of global eddies from satellite data by a scale-selective eddy identification algorithm (SEIA). Clim Dyn (2023). https://doi.org/10.1007/s00382-023-06946-w
> Global Oceanic Eddy Identification: A Deep Learning Method From Argo Profiles and Altimetry Data DOI=10.3389/fmars.2021.646926

Some published papers are also providing eddy mesoscale atlases at a global or regional level, others are on idealized eddy resolving model applied to North Pacific to identify rotationally coherent Lagrangian vortices (e.g. https://doi.org/10.1029/2019JC015576).

In practice I consider the article not original and based on algorithms consolidated in the literature. However, the work, done by a young researcher, is interesting, since it enriches an important theme.

Thank you for your comments. We respectfully disagree with the suggestion that the article is "not original". Certainly, our study benefits from the previously published literature, as we have acknowledged throughout the manuscript with extensive citations (now with an enhanced review in the revised Introduction section). However, it provides several novel aspects and presents an openly available atlas of coherent features that is distinctively suited for biogeochemical applications. Specifically, it uniquely provides tracking of coherent features (i) over their full lifetime, (ii) at high time

resolution, synchronized with remotely sensed chlorophyll products, and (iii) identified using backward Lagrangian trajectories because the recent history of coherency (rather than its future) is most valuable for interpreting associated biogeochemical signatures.

How is this atlas different from previously published Rotationally Coherent Lagrangian Vortex (RCLV) datasets? To our knowledge, there is currently one publicly available dataset of real-world RCLVs (Liu and Abernathy, 2023; doi.org/10.5194/essd-15-1765-2023). In that work, the authors interpreted RCLVs using Lagrangian particles initialized on the first day of every month and advected 180 days forward in time. They identified RCLVs that persisted for fixed intervals of 30, 90, or 180 days since the initializations. They do not track individual coherent eddies throughout their entire lifespan but rather capture the structures that happen to exist during those fixed intervals.

Here we describe the first methodology to detect and track all RCLVs within a domain *for the entirety of their individual lifespans*, creating the first RCLV "atlas". To do so, we closely overlap the initializations of the Lagrangian particle simulations and link features between the time steps based on particle trajectories. By performing the Lagrangian trajectory analysis in periods that overlap at 8-day intervals, we carefully synchronize our RCLV atlas with existing remotely sensed chlorophyll products (OC-CCI), enabling direct biophysical analysis and interpretation as illustrated in Figure 9 (now Fig. 11 of the revised manuscript). Finally, the RCLV dataset generated by Liu and Abernathy (2023) used forward-in-time Lagrangian trajectories and hence informs how long into the future a feature will remain coherent. In contrast, we used backward-in-time trajectories, thus revealing how long in the past a feature has been coherent, which is more valuable when seeking to interpret, for example, the origin of a local chlorophyll anomaly.

Thus, in our view, the manuscript and associated RCLV atlas have several original and uniquely useful features that were perhaps not articulated precisely enough in the original form. Thus, in the revised text, we sought to bring forward the novelty of the work more clearly. To this end, we added clarifying sentences outlining these distinctions in the revised Introduction (**Lines 108-109; 129-133**) and reformulated the abstract.

With regards to the specific literature cited by the reviewer above, these publications mostly concern improvements to Eulerian, altimetric-based eddy detection methods that are not sufficient to characterize the boundaries of coherency (see manuscript **Lines 71-80**). The Lagrangian study cited by the reviewer (Liu et al., 2019; https://doi.org/10.1029/2019JC015576) is based on a model simulation. The novel aspects of our study described above differentiate our manuscript from this work as well. We extended the literature review concerning the results of Liu et al. 2019 in the revision (**Lines 100-105**; see Specific Comments below).

Specific comments

The paper is presenting a general literature revue in &1.2, but from my point of view is insufficient.

In the revised manuscript, we expanded the literature review. For organizational clarity, we also separated the previous Section 1.2 into two sections: 1.2: "Eulerian eddy detection from satellite altimetry" and 1.3: "Lagrangian coherency detection from satellite altimetry". In particular, we substantially expanded the literature review of Lagrangian methods in Section 1.3 (**Lines 92-107**).

There is a plethora of Eulerian eddy identification algorithms including those listed by the reviewer. Thus, we have focused on the Eulerian SLA and OW eddy identification methods which have most often been utilized in studies of the biogeochemical response to eddies. Since we compared our RLCV atlas to an SLA atlas in this manuscript added a review of the recent developments in SLA eddy atlases in Section 1.2 (**Lines 63-74**).

It should include also a discussion on open questions and propose the solutions

Thank you for this suggestion. The manuscript now leads more naturally into the Results and Discussion because we added several open questions that are addressed by our analysis in the revised Introduction:

> *"Besides the direct applications for interpreting biogeochemical responses, there are several open questions that a Lagrangian coherent eddy atlas would address. Previous studies found that fewer and smaller structures maintain coherency for longer timescales (Abernathey and Haller, 2018; Xia et al., 2022b; Liu and Abernathey, 2023). However, the average lifespan of a coherent structure is unknown. How coherent structures change in size throughout their life is also unexplored. Furthermore, it is unclear whether eddies maintain the same coherent properties over their lifetimes or if they vary. For example, are nonoverlapping RCLVs always non-overlapping, or do they overlap with SLA eddies at some point? Is there a life stage where eddies are most likely to be coherent? Comparing RCLV and SLA eddy atlases can answer these questions, revealing how the coherent properties of mesoscale features manifest in space and time."* (**Lines 117-124**)

Since we highlight the evolution of RCLV size, we moved (and edited) what was Appendix Figure D2 to the main text, now Figure 7.

 e.g. (few questions coming in my mind)

We thank the reviewer for their suggested questions; here we individually address points raised in those:

SSH eddies overestimate coherent core and fail to reveal more than half of Lagrangian eddies: is the paper providing an answer?

*Previous works have also identified cases of "non-overlapping RCLVs", which we added to the literature review in the Introduction: "In an idealized model of the North Pacific, Liu et al. (2019) distinguished three types of eddy features: non-overlapping SLA eddies, nonoverlapping RCLVs, and overlapping eddies. Overlapping features are detected with both methods and non-overlapping are detected by one. RCLVs tend to be smaller and nested within the bounds of SLA eddies for overlapping features." (**Lines 101-103**)*

*We added a related open question to the reviewer's suggestion in the Introduction: ".. are nonoverlapping RCLVs always non-overlapping, or do they overlap with SLA eddies at some point?" (**Line 122**). By building an atlas, we reveal more information about non-overlapping RCLVs: "..  the majority of the occurrences of non-overlapping RCLVs are short-lived features, developing SLA eddies, or the remnants of SLA eddies" (**Line 450-451**)*

Looking at figures and movies, is there a significant difference between mesoscale eddies from random ocean pieces in material coherence?

*This is a great question that has been addressed in a previous study. We added the details to the Introduction: "Liu et al. (2019) found that the leakage rate of non-overlapping SLA eddies was higher than for overlapping eddies and that their dispersal properties were not significantly different from random ocean pieces of the same size." (**Lines 103-105**)*

How effective is transport, including chaotic stirring and filamentation outside of eddy cores?

*Eddy mass transport has been the focus of nearly all other studies that invoke the RCLV method. Abernathey and Haller 2018 (doi.org/10.1175/JPO-D-17-0102.1) & Xia et al. 2022 (doi.org/10.1175/JPO-D-21-0103.1) both make conclusions about the transport contributions of the coherent and leaky zones of eddies. These papers are cited in our literature review but for other aspects of their work. Though an interesting issue, since transport is not the focus of our paper (another distinction of our work from previous RCLV studies), we did not comment further in this manuscript.*

The definition of Lagrangian eddy must be made explicit. There are many possible choices, but in this paper is based on concept of coherent rotation of water parcels over the eddy lifetime (RCLV and LAVD are two alternatives, but here the second is adopted),

*We agree that there are other choices for Lagrangian coherent structure detection, and we described the FSLE/FTLE methods (**Lines 85-92**) since these have been most utilized in biogeochemical research. In the revised Introduction we cited a review paper on Lagrangian coherent structure detection and explicitly defined RCLVs as the choice of Lagrangian eddy that was previously defined in the Methods section. The RCLV is not an alternative method to the LAVD, as the reviewer suggests, rather RCLVs are identified directly from the LAVD: RCLVs are a subset of the high LAVD water parcels which also have rotational coherency. We have added the following new text:*

> *"Several methods exist for Lagrangian coherent eddy detection (reviewed by Hadjighasem et al., 2017). One of which is the Lagrangian Averaged Vorticity Deviation (LAVD), an observer-independent method that monitors the vorticity of fluid parcels to reveal Rotationally Coherent*

*Lagrangian Vortices (RCLVs; Haller et al., 2016). Here we define Lagrangian eddies (or RCLVs) using the LAVD method due to its objectivity, computational efficiency, and mathematical foundation describing material rotation."* (**Lines 93-97**)
* * *
**RC2: Peter Cornillon**

I really enjoyed this manuscript: I learned more from it than I generally do from manuscripts I review, it was well written and, I believe that the dataset they are introducing is of value, potentially significant value. Having said this, I do have a number of comments/suggestions, none of which relate to significant issues with the approach or results. These fall into two categories, simple editorial suggestions to improve clarity and, somewhat more significant comments, suggesting additional analyses or plots. I want to emphasize that these are suggestions for the authors to follow or not. Both sets of comments are indicated in the attached marked-up version of the manuscript, however, I will repeat the more significant ones here. (Oh yeah, I have also highlighted in yellow some of the definitions used. These, which the authors should ignore, were for my own benefit as I reviewed the manuscript.)

Thank you for your encouraging comments as well as the very helpful and detailed suggestions.

1) I'm a little confused about the definition of CD. My sense is that it should have dimensions of length: contour area / perimeter of the enclosing boundary but it is presented as unitless? I'm assuming that 'contour area' is the area enclosed by a closed contour?

We clarify the definition of CD in the revision: *"The CD is the ratio of the area enclosed by a contour to the area between the curve and its convex hull (Haller et al., 2016). The convex hull is the closed curve with the shortest perimeter containing the polygon. For example, a perfectly circular contour has CD = 0 because the convex hull is identical to the contour itself. Conversely, the area between a "wiggly" contour and its convex hull is higher and yields a high CD"* (**Lines 221-224**)

2) I struggled a bit with what was meant by overlap of RCLV over SLA or vice versa. It might help to label a few examples of each in Fig. 1 and to say, when introducing the idea, that rarely is an eddy of one species not almost completely contained in one of the other species. I have more comments related to this in the marked-up version of the manuscript.

We include the following related figure updates in the revision:

- Highlighted examples of "overlapping" and "non-overlapping" eddies in Figure 1
- Figure 5 was added to give a visual demonstration of an overlapping eddy and the relative retentiveness of the SLA and RCLV elements
- Added a figure to the Appendix (Figure D3) showing the area of overlap

We have also introduced the overlapping concept earlier, now in the Introduction: *"Comparisons of RCLVs and SLA eddies provide insight into the trapping properties of mesoscale features. In an idealized*

*model of the North Pacific, Liu et al. (2019) distinguished three types of eddy features: non-overlapping SLA eddies, nonoverlapping RCLVs, and overlapping eddies. Overlapping features are detected with both methods and non-overlapping are detected by one. RCLVs tend to be smaller and nested within the bounds of SLA eddies for overlapping features."* (**Lines 100-103**).

*In Section 3.2 of the revision we included more statistics on overlapping eddies: ".. for 33.3% of overlapping eddies, the RCLV is completely contained within an SLA eddy boundary. SLA boundaries are completely contained within an RCLV for 0.91% of the overlapping eddies cases. For the remaining overlapping eddies, the RCLV still tends to be mostly nested within the SLA, with an average of 0.22 of its relative area outside of the SLA eddy bounds (Appendix Fig. D3)."* (**Lines 342-345**)

3) It is argued that the area versus lifespan plot of RCLV eddies show a linear relationship while a similar plot for SLA eddies does not. First, the linearity of the relationship looks relatively weak but I'm willing to go along with it. I also agree that the SLA distribution is less linear. However, if one concentrates on the same area—lifespan range as that of the RCLV eddies, 0-3 km, 0-140 days the distribution looks like it may be close to that of RCLV, i.e., as linear, although it's hard to tell from the plot as is.

We maintain our claim that the relationship between RCLV area and lifespan is linear ($R^2$ = 0.6677). Note that in Figure 6, the color scale for the 2D histograms is lognormal. The RCLV cases that deviate from the linear relationship are rare compared to the dataset as a whole.

We agree that it is difficult to tell whether the distributions are more similar for smaller eddy sizes from Figure 6 alone. Therefore, we duplicated the plot while restricting the area to 40,000 $km^2$ and added this to the Appendix (Figure D4). We describe the results in the text: "*We also tested the relationship between the size and lifespan of the SLA eddies within a similar size range as the RCLVs (maximum area <= 40,000 km2 ). Here we found a weaker relationship (R2 = 0.377) between maximum area and lifespan compared to no size restriction (Appendix Fig. D4).*" (**Lines 349-351**).

4) Also related to Fig. 6 I think that it might be interesting to show a figure similar to this one but with the color of each bin being the fraction of eddies in the bin that are RCLV eddies. This would give a sense for the probability of overlap as a function of size and duration. I think that it would be informative but would have to see the plot to be sure. Just an idea.

The difficulty with the suggested plot is that SLA eddies and RCLVs do not necessarily overlap each other for the entirety of their lives (see Figure 4), can persist for differing lengths of time, and their sizes relative to each other can change dramatically throughout their lifetimes (see Figure 11). We tested some variations of the suggested figure but were unable to find a meaningful relationship, likely because of the wide variability in SLA eddy size.

Instead, we feel that the following statistics described in the methods are related to the reviewer's inquiry: "*RCLVs that do not overlap with an SLA eddy for their entire lifetime have significantly shorter lifespans (median 32 days) than their overlapping counterparts (median 72 days). Of the tracked RCLVs that do overlap with an SLA during their lifetime, 54% develop prior to the detection of the SLA eddy*

*boundary and 66% continue to propagate after the concurrent SLA eddy is no longer detectable. This suggests that the majority of the occurrences of non-overlapping RCLVs are short-lived features, developing SLA eddies, or the remnants of SLA eddies.*" (**Lines 447-451**)

5) I found the distributions of eddies shown in Appendix E, in support of the discussion of Fig. 8 to be really interesting. There is some discussion of this but I had to dig to sort things out. I'm guessing that a lot of readers will skip the appendices and, as a result, will miss this. It might be worth moving Figs. E1 and E2 into the body of the text and discussing in a little more detail.

We also found the eddy distributions to be interesting but chose to keep the plots in the Appendix in the revision to not clutter the main manuscript. However, we have provided more detail on these results in the revised main text. (**Lines 372-379**).

6) I found Fig. 9, showing the notion of trapped parcels, to be very helpful. However, I wondered while looking at the figure whether or not a parcel near the edge of the RCLV eddy would have remained in the eddy—the example shown is for a parcel near the center of the RCLV eddy. I guess that what I'm saying is that I would have been more convinced had a parcel been chosen near its edge.  I understand that parcels on the edge of these eddies may leak out of them due to turbulence but still, I would have used one closer to the edge.

We originally chose to show the trajectory of the center particle of the RCLV so that the eddy boundaries were easier to see. However, we understand the reviewer's concern and plotted the trajectory of a different particle closer to the edge of the RCLV.

Actually, an interesting plot would show the portion of the initial RCLV eddy for which, say, less than 5% leaked out in one color (maybe red), the portion for which less than 5% remained in the corresponding SLA contour in a second color  (maybe green) and the region between the two in gray. I also realize that this is just one example but such a plot would still convey a lot of information.

We added Figure 5 in the revision that shows the locations of all particles initialized in the example eddy of Figure 9. The new figure serves multiple purposes: 1) It visually demonstrates the retentiveness of RCLVs, and the lack thereof for SLA eddies; 2) Helps the reader to understand what we mean by overlapping eddy;  3) Aids in the interpretation of the chlorophyll case study in Section 3.4.

7) Also related to Fig. 9 you discuss the August feature but not the rather startling increase at the end of September.

We added a comment on the September chl increase in the revised text: " *A large-scale bloom developed in September (Video S2), the signatures of which were enhanced inside of the coherent structure, over a month past the detection of the SLA eddy.*" (**Lines 432-433**).

We do not think this increase is "startling", because the chl-a anomaly is within the range of anomalies observed throughout its lifetime. The slope of the change is more extreme, but that is because it is the first occurrence where the eddy was located in the middle of a large-scale bloom. However, the

signature of this bloom was enhanced inside of the RCLV. Video S2 aids in the interpretation of this anomaly.

A useful corollary plot would be one showing the area of the SLA eddy, the area of the RCLV eddy, the mean Chl value in SLA and the mean Chl value in RCLV all four lines as a function of time.

We updated Figure 9 to have a panel (e) with the eddy area through time.

Again, I really enjoyed the manuscript.

Hope that this helps.

Here is a listing of all comments in the attached marked-up manuscript.

Notes in 'essd-2023-425'

Notes in Document

'essd-2023-425':

Comment:       The definition of ephemeral is 'lasting for a very short time'. 'Short' is, of course, a subjective term but my sense is that it is much shorter than the length of time characteristic of many mesoscale eddies. I think that I would remove ephemeral and add 'and temporal scales of weeks to months' at the end of the sentence'. Or just remove it without adding a temporal scale at the end of the sentence since you mention the temporal scale in a few sentences. *(essd-2023-425, p.1)*

We removed the word ephemeral to avoid confusion.

Comment:       I'm a little confused about the definition of CD. My sense is that it should have dimensions of length: contour area / perimeter of the enclosing boundary but it is presented as unitless below? I'm assuming that 'contour area' is the area enclosed by a closed contour? *(essd-2023-425, p.7)*

See the response to Major Point #1 above.

Comment:   Does the intensity of the contours represent age or something else? *(essd-2023-425, p.9)*

We added, "*where the color corresponds with the eddy age (darkest = oldest)*" in the caption of Figure 3.

Comment:       It took me a bit to sort out what you are saying here. I think that the thing that I missed at first was that in virtually all cases an RCLV contour completely or almost completely contains an SLA contour or vice-versa when they are referencing the same feature. I had imagined cases where one could contain a significant region that was outside the other; e.g., eddies of the same size but overlapping by, say, 50%. In retrospect, I see that this is very unlikely since they are working with the same SLA field although there are some cases where the overlap is not complete, such as the cyclone at 21 N, 229 E. It would have helped had you labeled several examples in Fig. 1, examples of RCLV

overlapping SLA, SLA overlapping RCLV and non-overlapping examples. You might also comment on the rarity of cases where the identified eddies are clearly the same but have more than, say, 10% of the area that is non-overlapping. *(essd-2023-425, p.11)*

See the response to Major Point #2 above.

Comment: It looks to me like they decrease in size over the last 20% of their lifetimes versus the last 50%—the 70% box looks like the mirror image about 0 of the 30% box. *(essd-2023-425, p.13)*

We moved what was Appendix Figure D2 to the main text, now Figure 7, and included a subplot of the median relative change in the area so that the trend can be visualized more clearly. This statement was originally made based on the sign of the medians. We clarified in the text: "*We found that following the initial 32-day detection, RCLVs tend to grow for the next third of their lifespan, then maintain their area, and finally decrease in size during the last third of their life (Fig. 7).*" (**Lines 337-338**).

Comment: The contours in Fig. 6 seem to suggest that for SLA eddies similar in size to RCLV eddies (<~ 40,000 km^2) the size/lifespan relationship is much tighter than for the full range of SLA eddies; much closer to that of RCLV eddies. *(essd-2023-425, p.13)*

See the response to Major Point #3 above.

Comment: Tough to see the e^4 label on the y-axis. *(essd-2023-425, p.14)*

We adjusted the font size for the labels in Figure 6.

Comment: Should this be 'of anticyclonic SLA' *(essd-2023-425, p.15)*

We added the word "anticyclonic".

Comment: What's really striking is that the relative distributions of cyclonic eddies are very similar over the entire domain while the distribution of anticyclonic eddies is similar everywhere except in the region west of the islands (Figs. E1 and E2). Think that this is what you are trying to say but I had to dig to sort it out.

We added a more explicit statement: "*The polarity probability is similar between the datasets across the domain except to the west of the Islands, in the region of the Hawaiian Lee Eddies.*" (**Lines 367-368**)

Hmmm… I just looked at the cyclonic distribution a little more carefully and a striking feature of that distribution is the near total lack of RCLV cyclonic eddies in the region 195-200 E, 18-20 N. I think that there's more here that is worth discussing. *(essd-2023-425, p.15)*

We added: "*Cyclonic SLA eddies also occur in the Lee Eddy region dominated by anticyclonic RCLVs, with little to no cyclonic RCLVs in the same location (Appendix Fig. E2). This may be driven in part by large SLA bounds extending further south than the corresponding RCLVs. Cyclones are unable to maintain*

*coherency in the southern Lee Eddy region likely due to the background vorticity generated by the large-scale currents."* (**Lines 375-379**)

Comment:    Does it mean that a subset of anticyclonic Lee Eddies are not coherent or that virtually none of them are? *(essd-2023-425, p.15)*

We revised the text to state: *"Anticyclonic SLA eddies hugging the west of the islands were detected without any corresponding equivalent in the RCLV atlas. This suggests that anticyclones that propagate close to the Islands are rarely coherent."* (**Lines 371-372**)

Comment:    I wonder if the SLA distribution would be better defined if you constrained it to eddies <~40,000 km^2? I guess that this would be similar to what you find in Fig. E3. *(essd-2023-425, p.16)*

We do not wish to constrain the SLA eddies to an arbitrary maximum size cutoff because that would artificially eliminate real features from the analysis. Moreover, some features may pass above and below the threshold over their lifetimes, resulting in problems with eddy tracking.

We interpret that you are wondering if only the large SLA eddies are significantly different from the RCLVs? The added Appendix Figure D4 shows that SLA eddies in a similar size range as RCLVs still do not have the same size relationships with lifespan, and thus likely are not the same features (i.e., not overlapping).

Comment:    A more dramatic and convincing demonstration would have been two parcels very close to one another but with one leaving the region and the other remaining in the eddy. For example, the end point of a parcel that is near (but just inside of the 23 April RCLV boundary) the end point of the orange trajectory. The orange parcel that you picked is very close to the edge of the SLA boundary while the blue parcel is very near the center of the RCLV eddy. *(essd-2023-425, p.17)*

See the response to Major Point #6 above.

Comment:    I think that I would use 'of' instead of 'in'' here? *(essd-2023-425, p.17)*

Corrected in the revision.

Comment:    I assume that this region includes the region inside the boundary. Might it be a better measure of the background to exclude the region with the boundary, effectively an annulus although not perfectly circular? *(essd-2023-425, p.17)*

This is indeed how we made the calculation. We edited the text to say "*The chl-a anomaly is defined as the difference between the mean chl-a in the region of twice the radius of the eddy boundary (excluding the eddy) and the mean chl-a inside of the boundary.*" (**Lines 414-416**)

Comment: Yes, but wouldn't this be true of almost any closed convex—not that RCLV boundaries are necessarily convex but they are like close to being so—contour on the interior of the SLA boundary since the Chl concentration decreases away from the middle of the eddy? *(essd-2023-425, p.17)*

Chlorophyll anomalies typically occur as a monopole or dipole structure (e.g. see McGillicuddy 2016; doi: 10.1146/annurev-marine-010814-015606). In a dipole, chl anomalies are more concentrated on the edge of the eddy. A monopole is what is observed in Figure 9, where the anomaly is in the center of the eddy. Either eddy pumping or eddy trapping are thus responsible for these anomalies. Because the chl-a anomaly does not extend beyond the bounds of the RCLV, we believe that dilution limitation is playing a role here. If there was no eddy trapping (e.g. in a non-overlapping SLA eddy), the signature of eddy pumping might indeed decrease systematically from the eddy center, outwards to the SLA eddy bounds.

Comment: Are there times when this RCLV eddy is not completely inside the SLA boundary? If it is not completely contained maybe rewording as 'despite the fact that at most xx% of RCLV eddy lies outside of the SLA eddy'? I'm guessing that xx would be less than 5%, which would make your point here as well as providing an example of "overlap". Actually, I would find a simple plot of fraction outside, the xx above, versus time (panel e in the above) to be interesting unless—or maybe even if—it is zero most of the time. *(essd-2023-425, p.17)*

See the response to Major Point #2 above. We attempted to plot the percent of overlap versus RCLV age and found no meaningful relationship.

Comment: mid to late July? *(essd-2023-425, p.17)*

We changed the text to: "*The magnitude of the positive chl-a declined in the RCLV for several weeks until July when concentrations were lower than the large-scale bloom in surrounding waters.*" (**Lines 427-430**)

Comment: How did you get values for 7 and 15 April when it appears that you have not defined the RCLV eddy? Maybe I'm not uderstanding how the red line was calculated? *(essd-2023-425, p.18)*

We added a clarification in the caption of Figure 9, "T*he value of "SLA excluding RCLV" is equivalent to the "SLA" anomaly in April before the RCLV is detected.*"

Comment: You discuss the August feature but don't mention the rather startling anomaly peak on 8 October. I'm guessing that it relates to uncertainties in the measurements either of area or Chl concentration.

See response to Major Point #7 above.

A useful plot would be one showing the area of the SLA eddy and the RCLV eddy as a function of time along with the mean Chl values in the eddies. *(essd-2023-425, p.18)*

We included another subplot (e) in what is now Figure 11 to show the changes in area of the RCLV and the SLA eddy over the feature lifetime.

Comment:    Given the large spread in the scatter plot of Fig. 6, arguing that it represents a linear relationship is a bit of a stretch. Furthermore, it appears to me that SLA eddies in a similar size/duration range have a similar 'linear' relationship but I agree that it is hard to tell for sure from the plots.

See the response to Major Point #3 above.

Thinking a bit more about this it might be interesting to show a figure similar to Fig. 6 but with the color of each bin being the fraction of eddies in the bin that are RCLV eddies. This would give a sense of for the probability of overlap as a function of size and duration. I think that it would be informative but would have to see the plot to be sure. Just an idea. *(essd-2023-425, p.19)*

See the response to Major Point #4 above.

Comment:    You might want to expand acronyms the first time they appear in an Appendix, which is referenced prior to the definition of the acronym in the main text. *(essd-2023-425, p.22)*

In the text of Appendix A, we expanded the acronym and pointed to the definition of an RCLV.

Comment:    Why do you show this in percentage here but present in the text as a factor. Personally, I prefer factor but, regardless, I would be consistent—less mental gymnastics for the reader. *(essd-2023-425, p.27)*

We changed the Appendix D Figures to be presented as factors rather than percentages, consistent with the text.

---

## Author Response (AR2)

We thank Dr. Peter Cornillon for their kind comments and careful read of the manuscript. Each of their comments (copied directly below in black font) are addressed below, and we describe our corresponding changes to the manuscript.
* * *
**Referee #2: Peter Cornillon**

As with the previous version, I really enjoyed this manuscript—even more this time through now that I had a better understanding of it and time for my mind to assimilate the ideas. The authors have addressed all of my comments from the first pass. I have a few quite minor comments, which the authors may want to address.

1) In my first review, I indicated that I was a bit confused by the definition of CD. The authors have clarified the definition but now, it seems to me that they have it upside down. They say: "CD is the ratio of the area enclosed by a contour to the area between the curve and its convex hull". They go on to say that "a perfectly circular contour has CD = 0 because the convex hull is identical to the contour itself." Given their definition, wouldn't that mean that CD is infinite, since the area between the curve and the convex hull is zero and, since CD is the ratio of the area in the contour to (which I take to mean divided by) the the area between the curve and the convex hull?

You are correct and thank you for catching that. We fixed the sentence in the revision: "*The CD is the ratio of the area between the contour and its convex hull to the area enclosed by the contour*" (Line 221).

2) Lines 231-232 "A low CI indicates that the particle set has spread apart over the time frame, whereas a high CI indicates compressing behavior." I always get confused when low and hi are used if the number can change sign; i.e., does it mean signed or magnitude. I realize that in this case they mean signed but it took me a rereading to sort this out. A simple rewording would address this "Negative CI indicates that the particle set is spreading and positive CI means that it is contracting."

We agree that your suggestion is clearer. We revised the sentence to: "*A negative CI indicates that the particle set spread apart over the time frame, whereas a positive CI indicates that it contracted*" (Lines 231-232).

3) Fig. 2, frame b, Is the label for CD correct? I'm guessing that it should be 0.019 or larger. But, it's tough to guess at this. If I'm wrong, no need to modify the text.

The label is correct in the figure. However, we understand how the figure labels might have caused confusion. Although the "target" CD was 0.03, the contour that was output by the floater algorithm has an "actual" CD of 0.119. The CD value is high due to the filament that it

encompasses in the lower right (see the green contour in panel e). This large difference in the "target" and "actual" CD occurred because the floater algorithm did not identify a contour close to the target. Instead, it output a contour located in between the contours with CDs incrementally lower and higher than the target, regardless of the actual CD.

More information on the decision tree for these boundaries is in Appendix B. We updated this text to be explicitly clear that the actual CD of a contour is not necessarily equivalent to the target CD (i.e. the input in for the algorithm):

> *"Starting with target CD = 0.03, we check if the outputted contour has an actual CD <= 0.03. In some cases, the actual CD may be much higher than the target if the algorithm cannot find a contour with the desired CD. If the actual CD passes the criterion, then…"* (Lines 559-561).

To avoid confusing the reader in the main text, instead of labeling the contours in panel (e) with the "target" CDs, we labelled them with the actual CDs of the contours. We also updated the caption of Figure 2 accordingly:

> *"An RCLV vortex and its potential boundaries derived by varying the target CD in the floater algorithm. (a) Particles were initialized inside the contour with CD = 0.011 and advected backward in time for 32 days. The particles are yellow at the initialization, and orange at their locations after advection. (b) Particles initialized in the contour with CD = 0.119. (c) Particles initialized in the contour with CD = 0.049. (d) Particles initialized in the contour with CD = 0.07. (e) Contour output from different target CDs."*

4) Line 356, when referring to the "eddies are in the Lee of the Hawaiian Islands" you might want to add parenthetically that the prevailing winds in this region are from the east northeast. Most people reading the manuscript will likely know this but for those who don't…

We added a sentence near the line suggested:

> *"The highest frequencies of both RCLVs and SLA eddies are in the Lee of the Hawaiian Islands. Eddies there are sustained by the background currents (Calil et al., 2008; Yoshida et al., 2010; Liu et al., 2012) and the wind stress curl generated where mountains interfere with the northeasterly trade winds (Lumpkin, 1998; Yoshida et al., 2010)."* (Lines 356-358).

We had detailed the generation mechanisms of the Hawaiian Lee Eddies in the Discussion section (Lines 469-475), rather than in the Results sections. There we also added the word "northeasterly" to describe the direction of the trade wind (Line 473).